# Female-germline specific protein Sakura interacts with Otu and is crucial for germline stem cell renewal and differentiation and oogenesis

**Azali Azlan, Li Zhu, Ryuya Fukunaga***

Department of Cell Biology Johns Hopkins University School of Medicine, Baltimore, United States

## eLife Assessment

This **valuable** study reports the first characterization of the CG14545 gene in *Drosophila melanogaster*, which the authors name "Sakura." Acting during germline stem cell fate and differentiation, Sakura is required for both oogenesis and female fertility, although some mechanistic details require further investigation. This **solid** study presents a wide-ranging and well-controlled characterization of Sakura, and accordingly the findings and associated reagents described will be of use to scientists interested in oogenesis and early development.

**\*For correspondence:**
fukunaga@jhmi.edu

**Competing interest:** The authors declare that no competing interests exist.

## Abstract

During oogenesis, self-renewal and differentiation of germline stem cells (GSCs) must be tightly regulated. The *Drosophila* female germline serves as an excellent model for studying these regulatory mechanisms. Here, we report that a previously uncharacterized gene *CG14545*, which we named *sakura*, is essential for oogenesis and female fertility in *Drosophila*. Sakura is predominantly expressed in the ovaries, particularly in the germline cells, including GSCs. sakura null mutant female flies display rudimentary ovaries with germline-less and tumorous phenotypes, fail to produce eggs, and are completely sterile. The germline-specific depletion of sakura impairs Dpp/BMP signaling, leading to aberrant *bag-of-marbles* (*bam*) expression, resulting in faulty differentiation and loss of GSCs. sakura is also necessary for normal levels of piwi-interacting RNAs (piRNAs) levels and for female-specific splicing of *sex-lethal* (*sxl*), a master regulator of sex identity determination. We identified Ovarian Tumor (Otu) as a protein binding partner of Sakura and found that loss of *otu* phenocopies loss of *sakura* in ovaries. Thus, we identify Sakura as a crucial factor for GSC renewal and differentiation and oogenesis, and propose that Sakura and Otu function together in these processes.

## Introduction

Oogenesis is a complex process in which GSCs develop into mature female gametes or oocytes. This process involves multiple layers of regulations and numerous genes, some of which remain to be discovered and characterized. *Drosophila melanogaster*, a genetically tractable model organism, has long been at the forefront of GSC and oogenesis research (*Kirilly and Xie, 2007*).

Adult *Drosophila* females possess a pair of ovaries, each composed of 12–16 ovarioles. At the anterior tip of each ovariole is a structure called the germarium, which houses two to three GSCs (*Figure 1A*). GSCs reside in the most anterior region of the germarium, where they directly contact cap cells and escort cells, forming a niche crucial for GSC self-renewal (*Lin and Spradling, 1993*;

*de Cuevas and Matunis, 2011*; *Losick et al., 2011*). GSCs undergo asymmetric mitotic division, producing two distinct cells: one GSC and one cystoblast. The cystoblast, destined to differentiate, undergoes four mitotic divisions with incomplete cytokinesis, resulting in interconnected cysts of 2, 4, 8, and 16 cells. Of the 16 cells, one will differentiate into an oocyte, while the remaining 15 develop into polyploid nurse cells, which synthesize proteins and RNAs for deposition into the oocyte.

GSCs must be tightly regulated to maintain their undifferentiated state while dividing and differentiating to produce one GSC and one cystoblast. Failure in GSC self-renewal or inappropriate differentiation leads to stem cell loss, disrupting oocyte production (*Lin, 1997*; *Cox et al., 1998*). Conversely, uncontrolled GSC division or defective differentiation results in an overabundance of GSC-like cells (a tumorous phenotype) and fewer differentiated cells, ultimately inhibiting oocyte production (*Gateff et al., 1996*; *Ohlstein et al., 2000*). Thus, a precise balance between GSC self-renewal and cyst differentiation within the germarium is essential for proper oogenesis and is tightly regulated.

Bone Morphogenetic Protein (BMP) signaling plays a critical role in regulating GSC self-renewal and differentiation during *Drosophila* oogenesis. In the stem cell niche, GSCs are physically anchored to cap cells via adherens and gap junctions (*Song et al., 2002*; *Gilboa et al., 2003*). Cap cells secrete BMP ligands, such as Decapentaplegic (Dpp) and Glass bottom boat (Gbb), which are recognized by transducing receptors like Thickvein (Tkv) and Saxophone (Sax) on the GSCs (*Xie and Spradling, 1998*; *Casanueva and Ferguson, 2004*). Upon activation, these receptors phosphorylate a transcription factor, Mad. The phosphorylated Mad (pMad) translocates into the nucleus, where it represses the transcription of *bam*, a key differentiation factor (*Kai and Spradling, 2003*; *Song et al., 2004*).

Bam, a ubiquitin-associated protein, is essential for cystoblast differentiation (*McKearin and Spradling, 1990*; *McKearin and Ohlstein, 1995*). Dpp/BMP signaling in the niche represses *bam* expression, preventing GSCs from differentiating and enabling self-renew (*Song et al., 2004*). However, as the daughter cells (cytoblasts) exit the niche, they derepress *bam* and initiate the differentiation program. Tight regulation of Dpp/BMP signaling and *bam* expression ensures that only cells within the niche adopt GSC fate, while adjacent daughter cells differentiate into cystoblasts. Loss of *bam* leads to blocked differentiation and the accumulation of GSC-like cells (tumorous phenotype), whereas ectopic *bam* expression forces GSCs differentiation, depleting the stem cell pool (*McKearin and Spradling, 1990*; *McKearin and Ohlstein, 1995*; *Ohlstein and McKearin, 1997*).

One of the proteins that interacts with Bam is Otu (*Ji et al., 2017*). Otu forms a deubiquitinase complex with Bam, which deubiquitinates Cyclin A (CycA), stabilizing CycA and promoting GSC differentiation (*Ji et al., 2017*). Otu also binds RNAs (*Ji et al., 2019*). Otu is crucial for oogenesis and female fertility. Mutations in the *otu* gene cause various ovarian defects, including tumorous growths, germ cell loss, abnormalities in oocyte determination and nurse cell dumping, and defects in the sexual identity determination of germ cells, indicating Otu's importance in multiple stages of oogenesis (*Smith and King, 1966*; *Gans et al., 1975*; *Gollin and King, 1981*; *King and Riley, 1982*; *Storto and King, 1988*; *Pauli et al., 1993*; *Rodesch et al., 1997*; *Glenn and Searles, 2001*). However, the precise molecular mechanisms by which Otu functions in these processes remain largely unknown.

In this study, we aimed to identify a novel gene required for oogenesis. Through a search for uncharacterized genes expressed exclusively in *Drosophila* ovaries, we discovered CG14545, a gene encoding a 114 amino acids (aa) long protein without any known functional domain (*Figure 1B*). We found that CG14545 is expressed specifically in female germline cells, including GSCs. Our genetic analysis revealed that CG14545 is required for oogenesis and female fertility and plays an important intrinsic role in regulating GSCs renewal and differentiation. Additionally, CG14545 is important for the piRNA pathway and for the female-specific mRNA splicing pattern of *sex-lethal* (*sxl*), a master regulator of sex determination. We identified Otu as a binding partner of the CG14545 protein, suggesting that both proteins may participate in the same molecular pathways to regulate GSC fate and differentiation. We named the CG14545 gene *sakura*, meaning 'cherry blossom' in Japanese, symbolizing birth and renewal.

## Results

### Sakura is exclusively expressed in the ovaries

Based on high-throughput expression data in Flybase, *sakura* mRNA is highly and exclusively expressed in adult ovaries (*Figure 1—figure supplement 1*). To investigate Sakura protein expression, we

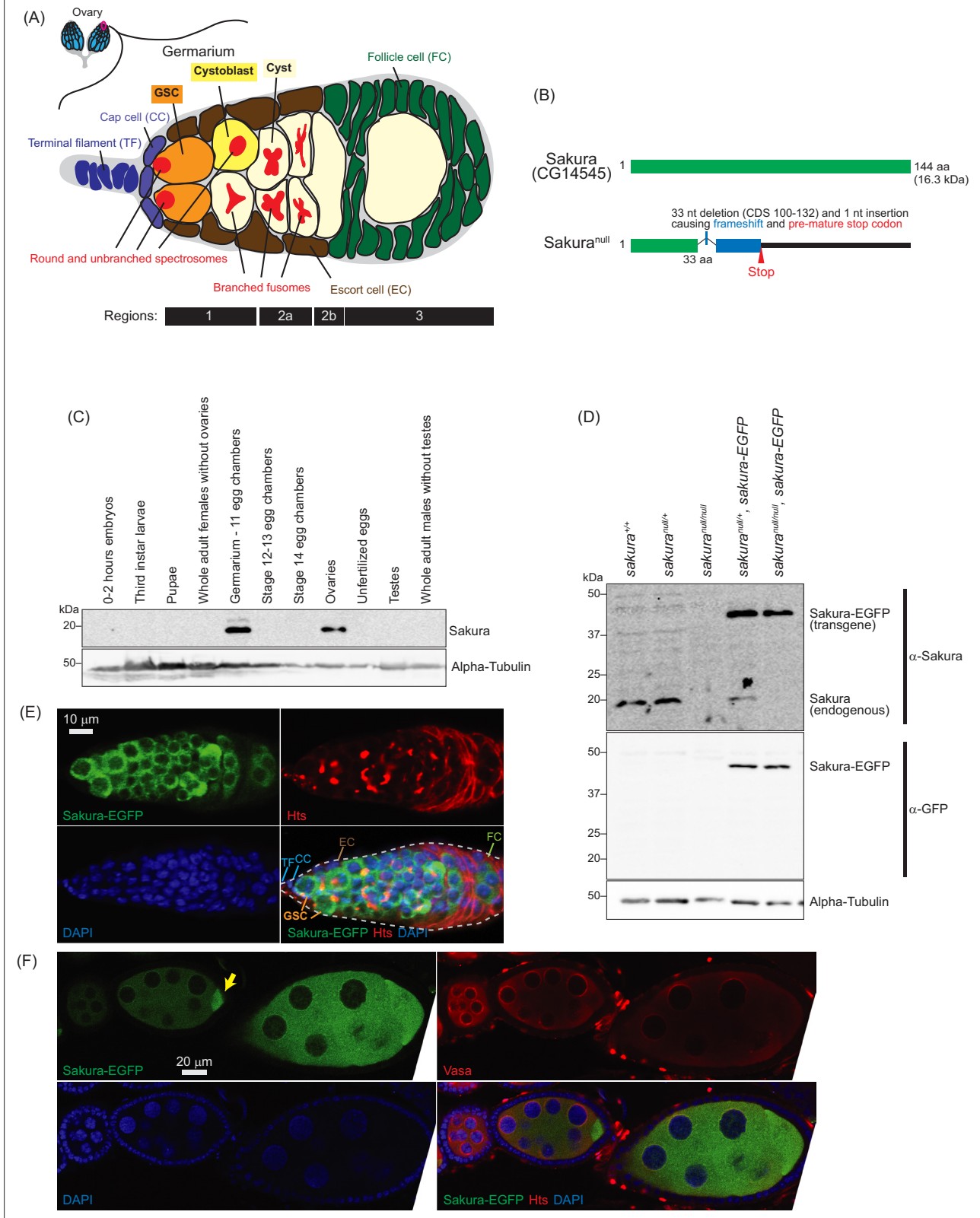

**Figure 1.** Sakura expression pattern and mutant allele. (**A**) Schematic illustration of *Drosophila* ovary and germarium. *Drosophila* female has a pair of ovaries, each consisting of 12–16 ovarioles (cyan). The germarium (outlined with magenta) is located at the anterior tip of the ovarioles and consists of both germ cells and somatic cells. Germ cells include germline stem cells (GSCs), cystoblasts, cysts, and differentiating oocytes. Somatic cells include terminal filament (TF) cells, cap cells (CCc), escort cells (ECs), and follicle cells (FCs). GSCs and cystoblasts have spherical, unbranched

*Figure 1 continued*

spectrosomes, whereas cysts possess branched fusomes. The distinct regions of the germarium—1, 2 a, 2b, and 3—are indicated. (**B**) *Drosophila* Sakura protein (Sakura/CG14545) and its null mutant allele were generated in this study. (**C**) Western blot of dissected fly tissues. (**D**) Western blot of ovary lysates. (**E**) Confocal images of the germarium from the *sakura-EGFP* transgenic fly. Sakura-EGFP (green), Hts (red), and DAPI (blue). Scale bar: 10 µm. (**F**) Confocal images of the egg chambers from the *sakura-EGFP* transgenic fly. Sakura-EGFP (green), Vasa (red), and DAPI (blue). Sakura-EGFP is expressed in nurse cells and enriched in the developing oocyte (yellow arrow). Scale bar: 20 µm.

The online version of this article includes the following source data and figure supplement(s) for figure 1:

**Source data 1.** Original uncropped gel blot images used in *Figure 1C and D*, indicating the relevant bands.

**Source data 2.** Original uncropped, unedited gel blot image files used in *Figure 1C and D*.

**Figure supplement 1.** *sakura* mRNA expression pattern.

generated a polyclonal anti-Sakura antibody against a recombinant full-length Sakura protein. Using this antibody, we examine Sakura protein expression at different stages of *Drosophila* development and across tissues. We found that Sakura protein is specifically expressed in ovaries, particularly in germarium - 11 egg chambers (*Figure 1C*).

### *sakura* mutant flies

To explore the biological and molecular functions of Sakura in vivo, we generated a *sakura* mutant allele (*sakura^null^*) with a one nucleotide (nt) insertion replacing 33 nts within the Sakura coding region using the CRISPR/Cas9 genome editing system (*Figure 1B*). This mutation causes a frameshift and a premature stop codon, resulting in a 33-amino acid (aa) N-terminal fragment of Sakura followed by a 22-aa segment produced by frameshifted translation (*Figure 1B*). This truncated fragment is unlikely to be functional, and we were unable to detect stable protein expression, as shown below (*Figure 1D*). Thus, we consider this allele a null mutation. *sakura^null/null^* flies were viable, revealing that Sakura is not essential for survival.

To validate the *sakura^null^* strain, we performed Western blots using ovary lysates from *sakura* mutant flies using the anti-Sakura antibody. As expected, full-length Sakura protein was detected in the ovaries of wild-type (*sakura^+/+^*) and heterozygous controls (*sakura^null/+^*), but not in those of homozygous mutants (*sakura^null/null^*) (*Figure 1D*). No smaller protein corresponding to the truncated Sakura fragment was detected in either *sakura^null/+^* or *sakura^null/null^* ovaries, thus suggesting that the fragment is unstable or not expressed.

### Sakura is cytoplasmic and expressed specifically in germ cells

To determine the localization of Sakura protein within the ovaries, we generated transgenic flies carrying a Sakura coding sequence fused with EGFP at the C-terminus (Sakura-EGFP) under the control of the *sakura* promoter. Western blot analysis of ovary lysates showed that Sakura-EGFP is expressed at levels comparable to endogenous Sakura (*Figure 1D*).

Oogenesis begins in the germarium, which contains 2–3 GSCs identified by round, unbranched spectrosomes that contact cap cells (*Figure 1A*; *Kirilly and Xie, 2007*). In contrast, cysts exhibit branched fusomes. Immunostaining with hu-li tai shao (HTS) antibody marks both spectrosomes and fusomes (*Figure 1E*), while Vasa is a known germ cell marker (*Figure 1F*). Confocal imaging revealed that Sakura-EGFP is specifically expressed in the cytoplasm of germ cells, including GSCs, cyst cells, nurse cells, and developing oocytes (*Figure 1E and F*). Sakura-EGFP was not expressed in somatic cells such as terminal filaments, cap cells, escort cells, or follicle cells (*Figure 1E*). In the egg chamber, Sakura-EGFP was detected in the cytoplasm of nurse cells and was enriched in developing oocytes (*Figure 1F*).

### *sakura* is essential for female fertility

Given its ovary-specific expression at both mRNA and protein levels, we hypothesized that Sakura plays a critical role in ovarian function. Fertility assays revealed that *sakura^null/null^* females laid no eggs when mated with wild-type males, while control females (*sakura^+/+^* and *sakura^null/+^*) laid numerous eggs (*Figure 2A and B*). Dissection of *sakura^null/null^* ovaries revealed that they were rudimentary compared to normal ovaries in controls (*Figure 2C*). This suggests that the inability of *sakura^null/null^* female to lay eggs is due to their underdeveloped ovaries. In contrast, *sakura^null/null^* males were fertile and showed

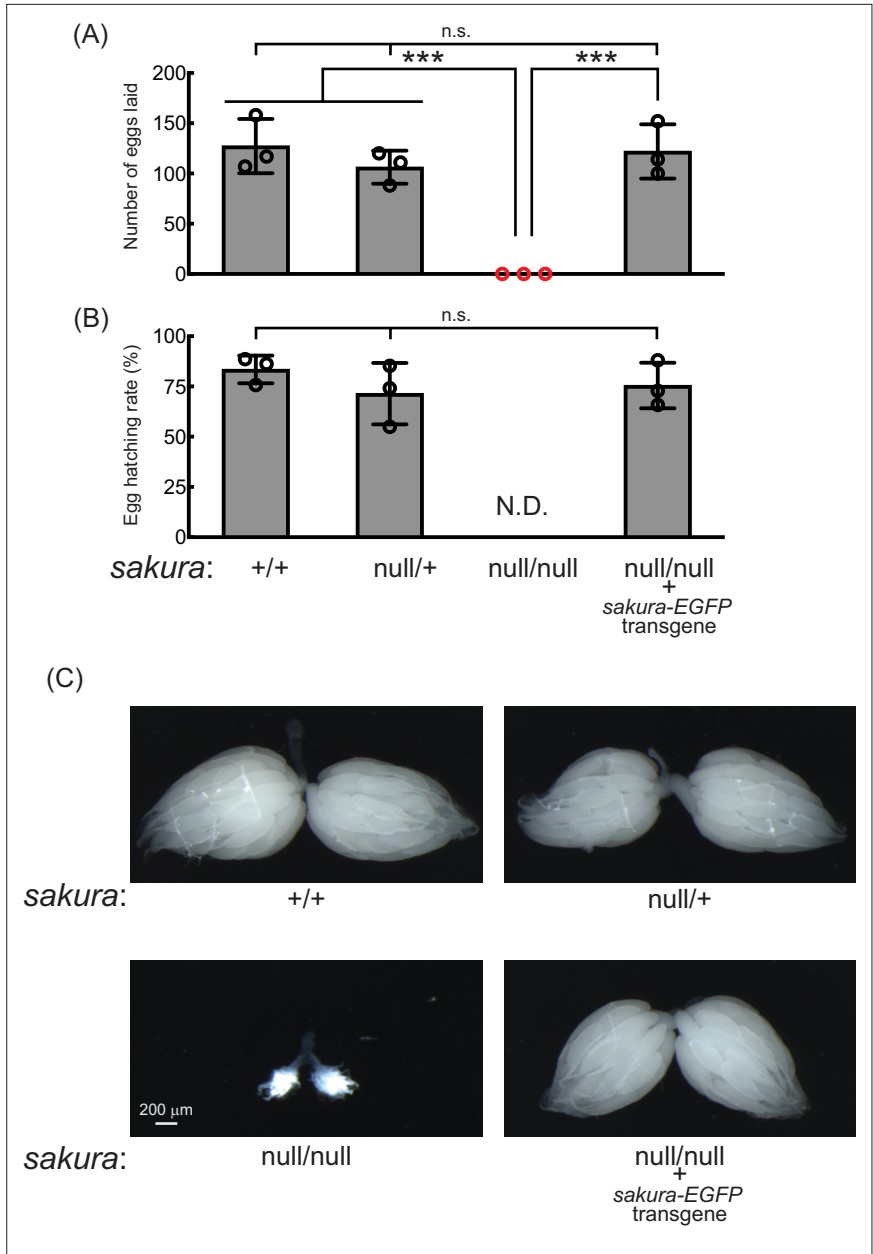

**Figure 2.** *sakura* mutant flies are female-sterile and have rudimentary ovaries. (**A–B**) Female fertility assays. (**A**) The number of eggs laid by test females crossed with OregonR wild-type males and (**B**) hatching rates of the eggs. Mean ± SD (n = 3). P-value < 0.001 (Student's t-test, unpaired, two-tailed) is indicated by \*\*\*. (**C**) Stereomicroscope images of dissected whole ovaries. Scale bar: 200 μm.

The online version of this article includes the following figure supplement(s) for figure 2:

**Figure supplement 1.** Male fertility assay.

no significant differences compared to controls (*sakura^{+/+}* and *sakura^{null/+}*) (*Figure 2—figure supplement 1*).

To confirm that the observed female sterility was caused by the loss of *sakura*, we generated *sakura* rescue flies expressing Sakura-EGFP in the *sakura^{null/null}* background (*sakura^{null/null}*; *sakura*-EGFP). Western blotting confirmed that these flies expressed Sakura-EGFP, but not endogenous Sakura (*Figure 1D*). The *sakura*-EGFP transgene fully rescued both female fertility and ovary morphology in the *sakura^{null/null}* background (*Figure 2*). *sakura*-EGFP rescue females showed no significant fertility differences compared to controls (*Figure 2A and B*), and their ovaries were normal in shape and size

(*Figure 2C*). We concluded that Sakura is essential for female fertility and normal ovary morphology but dispensable for male fertility, consistent with the observation that Sakura is predominantly expressed in ovarian germline cells (*Figure 1*).

## *sakura*<sup>null/null</sup> ovaries are germless and tumorous

Given the rudimentary appearance of *sakura*<sup>null/null</sup> ovaries, we questioned whether they contained germ cells. We used flies carrying vasa-EGFP knocked-in reporter in the *sakura*<sup>null/null</sup> background, as Vasa is a known germ cell marker. We observed that some *sakura*<sup>null/null</sup> ovarioles were devoid of germ cells ('germless,' cyan stars), while others retained germ cells (*Figure 3A and B*). Immunostaining with HTS antibody revealed an excess number of GSC-like cells with round spectrosomes in *sakura*<sup>null/null</sup> ovarioles containing germ cells (*Figure 3C*, orange stars), indicative of a 'tumorous' phenotype as previously described for mutants of other genes, including *bam, otu, and sxl* (*Smith and King, 1966*; *Gans et al., 1975*; *Gollin and King, 1981*; *King and Riley, 1982*; *McKearin and Spradling, 1990*; *Bopp et al., 1993*; *Eliazer et al., 2011*; *Jin et al., 2013*; *Yang et al., 2019*). Additionally, we observed an excess number of cyst cells with branched fusomes that persisted throughout the ovarioles, suggesting abnormal cyst cell differentiation and division. Thus, the loss of *sakura* results in both germ cell depletion and overgrowth.

Within the same ovary, some *sakura*<sup>null/null</sup> ovarioles exhibited the tumorous phenotype, while others were germless, without any spectrosome or fusome staining (*Figure 3A–C*, cyan stars). Quantification revealed that 35% of *sakura*<sup>null/null</sup> ovarioles from 2–5-day-old flies were tumorous, while the remaining 65% were germless (n=74) (*Figure 3D*). In contrast, all ovarioles in control (*sakura*<sup>+/+</sup> and *sakura*<sup>null/+</sup>) and *sakura-EGFP* rescue flies were normal with no tumorous or germless phenotypes observed (*Figure 3D*). Approximately 95% of *sakura*<sup>null/null</sup> ovarioles containing germ cells had an excess of GSC-like cells (>5) (*Figure 3E* and *Figure 3—figure supplement 1*). The mean number of GSCs or GSC-like cells in 2–5-day-old *sakura*<sup>+/+</sup>, *sakura*<sup>null/+</sup>, and *sakura-EGFP* rescue was 2.2±0.6, 2.2±0.6, and 2.1±0.7, respectively while that for *sakura*<sup>null/null</sup> was 26.5±17.5 (p-value<0.05).

We investigated the age-dependent effects on the tumorous and germless phenotypes by examining the ovarioles from 0 to 1 day old, 7 day old, and 14 day old flies. Ovarioles in control and *sakura-EGFP* rescue flies remained normal throughout this time course (*Figure 3—figure supplement 2*). In the very young, 0–1 day old *sakura*<sup>null/null</sup>, 78% of ovarioles were tumorous and 22% were germless. Overtime, the ratio of germless ovarioles increased while the ratio of tumorous ovarioles decreased. By 14 days old, only 5% of ovarioles were tumorous, with the remaining 95% being germless. The number of GSC-like cells in *sakura*<sup>null/null</sup> ovarioles was already high in 0–1 day old flies, and it decreased over time (*Figure 3—figure supplement 2*). Tumorous ovarioles, but not germless ones, exhibited significantly higher levels of cleaved Caspase-3 staining, indicative of apoptosis, compared to controls (*Figure 3F*). These results suggest that tumorous ovarioles undergo apoptosis, and that these ovarioles eventually become germless.

Together, these results suggest that Sakura is essential for regulating the survival, proliferation, and differentiation of germline cells, including GSCs.

## Loss of *sakura* results in reduced germline piRNA

In controls, Vasa-EGFP is enriched in the perinuclear structure known as the nuage within nurse cells (*Figure 3A and B*), which is essential for piwi-interacting RNA (piRNA) biogenesis. In *sakura*<sup>null/null</sup> tumorous ovarioles, Vasa-EGFP retains its perinuclear localization, indicating that *sakura* is not required for the proper localization of Vasa to the nuage.

piRNAs are produced in germline cells and somatic follicle cells and transcriptionally and post-transcriptionally silence transposon expression through sequence-complementarity (*Huang et al., 2017*; *Yamashiro and Siomi, 2018*). Disruption of the piRNA pathway can result in oogenesis arrest, germ cell loss, rudimentary ovaries, and sterility. Loss of piRNAs in germ cells leads to the upregulation (desilencing) of transposons, increased DNA damage, and apoptotic cell death. We investigated whether the loss of *sakura* would result in the loss of piRNA and transposon upregulation. We performed high-throughput sequencing (small RNA-seq and polyA +RNA seq) on ovary RNA samples from *sakura*<sup>+/+</sup>, *sakura*<sup>null/+</sup>, *sakura*<sup>null/null</sup>, and *sakura-EGFP* rescue (*sakura*<sup>null/null</sup> with *sakura-EGFP*) flies.

piRNA levels in *sakura*<sup>null/null</sup> ovaries were reduced compared to controls (*sakura*<sup>+/+</sup>), while *sakura*<sup>null/+</sup> and *sakura*-EGFP rescue showed no such reduction (*Figure 4A*). Consistent with this, transposon RNA

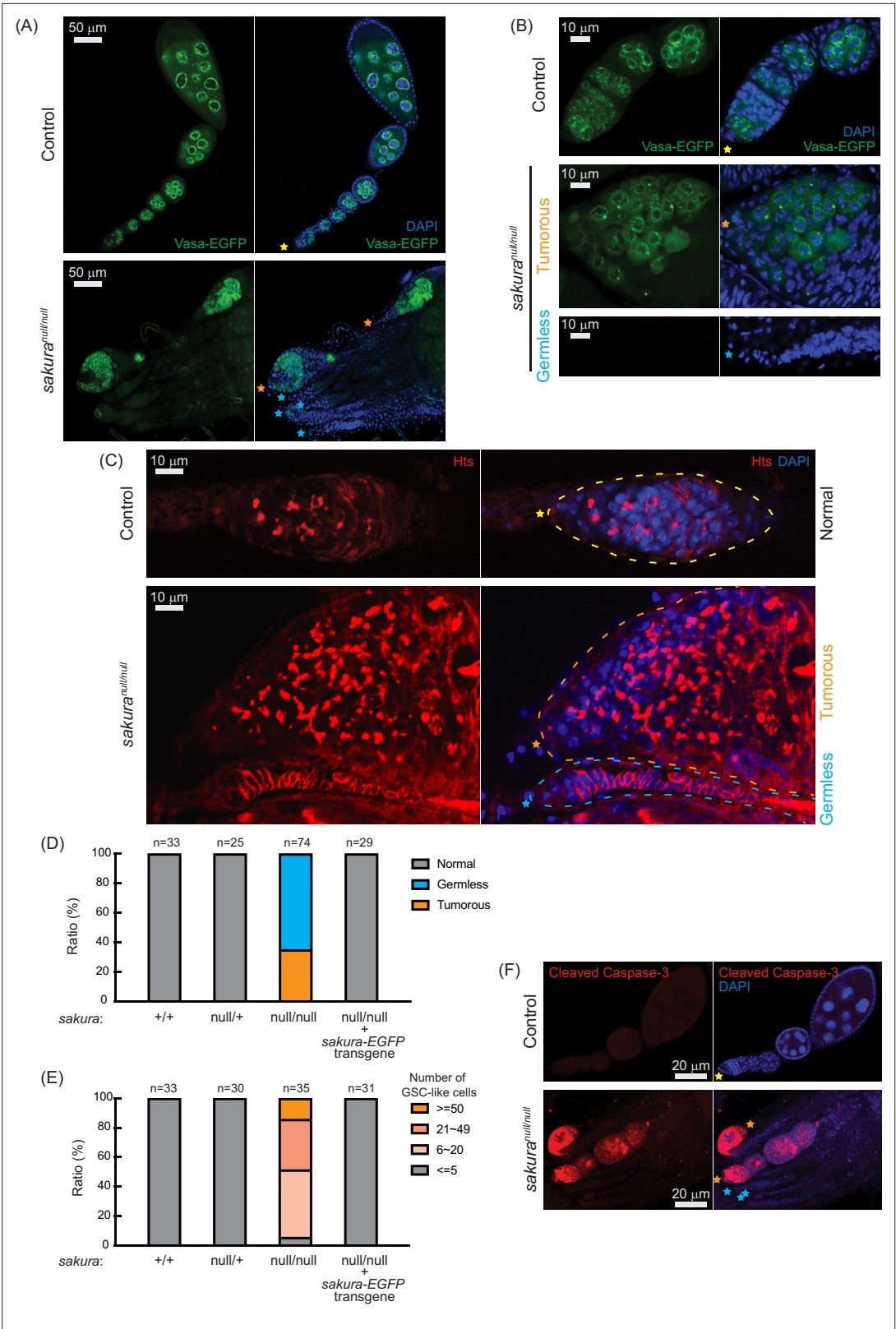

**Figure 3.** *sakura^null* ovaries are germless and tumorous. (**A, B**) Confocal images of the ovaries from control (*sakura^null/+*) and *sakura^null/null* expressing Vasa-EGFP. Vasa-EGFP (green) and DAPI (blue). Yellow stars indicate the anterior tip of normal ovarioles, while orange and cyan stars indicate the anterior tips of tumorous and germless ovarioles, respectively, in Figure 3. Higher-magnification images of the germarium regions are shown in (**B**). Scale bars: 50 µm for (**A**) and 10 µm for (**B**). (**C**) Confocal images of control (*sakura^null/+*) and *sakura^null/null* ovaries stained with anti-Hts antibody to label spectrosomes

*Figure 3 continued on next page*

Figure 3 continued

and fusomes. Hts (red) and DAPI (blue). Yellow, orange, and cyan dotted lines mark the normal, tumorous, and germless germaria, respectively. Scale bars: 10 µm. (**D**) Ratio (%) of normal, germless, and tumorous ovarioles of indicated genotypes (ages 2–5 days; n=33, 25, 74, 29, respectively). (**E**) Quantification of germline stem cell (GSC)-like cell number per germarium in the indicated genotypes (ages 2–5 days; n=33, 30, 35, and 31 respectively). (**F**) Confocal images of control (*sakura^null/+*) and *sakura^null/null* ovaries stained with anti-cleaved Caspase-3 antibody. Cleaved caspase-3 (red) and DAPI (blue). Scale bars: 20 µm.

The online version of this article includes the following figure supplement(s) for figure 3:

**Figure supplement 1.** *sakura^null* ovaries are tumorous.

**Figure supplement 2.** The ratio of germless ovarioles increases over time in *sakura^null* ovaries.

---

levels in *sakura^null/null* ovaries were upregulated compared to *sakura^+/+*, while *sakura^null/+* and *sakura-EGFP* rescue showed no such upregulation (*Figure 4B*). Given that Sakura is specifically expressed in germline cells, but not in somatic cells (*Figure 1E and F*), we hypothesized that loss of *sakura* interrupts the piRNA pathway in germ cells, leading to the desilencing of germline transposons. A well-characterized germline transposon in *Drosophila* is Burdock (*Dönertas et al., 2013*; *Handler et al., 2013*). *We found that Burdock piRNA levels were significantly lower in sakura^null/null compared to controls and rescue flies (Figure 4C).*

We employed the Burdock sensor, a transposon reporter tool to monitor germline piRNA activity (*Handler et al., 2013*). The reporter expresses nuclear GFP and β-gal under the control of the nanos promoter for germline expression, with a target sequence for Burdock piRNAs in the 3′ UTR (*Figure 4D*). Using a *sakura* RNAi line driven by *UAS-Dcr2* and NGT-Gal4 to knock down *sakura* specifically in the germline, we observed that GFP and β-gal expression were highly elevated in sakura knockdown germlines compared to control RNAi (y^RNAi) knockdowns (*Figure 4D*). This confirmed that the loss of *sakura* leads to a loss of piRNA-mediated transposon silencing in the germline, suggesting that Sakura is essential for proper piRNA levels and piRNA-based transposon silencing in the germline.

## Sex-specific *sxl* mRNA alternative splicing is dysregulated in *sakura^null/null* ovaries

Sxl is a master regulator of sex determination in *Drosophila* (*Penalva and Sánchez, 2003*; *Salz and Erickson, 2010*; *Grmai et al., 2022*). Sex-specific alternative splicing of *sxl* transcripts produces distinct mRNA isoforms: the female-specific and male-specific mRNA isoforms in respective sex and only the female-specific mRNA isoform encodes the functional Sxl protein, whereas the male-specific isoform does not. In the female germline, loss of Sxl function or disruption of female-specific *sxl* splicing leads to developmental defects, including germline tumors and sterility. Notably, several mutants with tumorous ovariole phenotypes—including *otu* mutants—exhibit aberrant *sxl* mRNA splicing, resulting in the expression of the male-specific isoform in ovaries and defects in germ cell sexual identity (*Bopp et al., 1993*; *Pauli et al., 1993*).

We examined *sxl* splicing in *sakura* mutants. As expected, ovaries from control and *sakura-EGFP* rescue flies expressed exclusively the female-specific *sxl* mRNA isoform and testes from control flies expressed only the male-specific isoform (*Figure 4—figure supplement 1*). In contrast, *sakura^null/null* ovaries exhibited the male-specific isoform and a reduced level of the female-specific isoform. These results indicate that female-specific *sxl* alternative splicing is disrupted in the absence of *sakura*.

## *sakura* is important for oogenesis in germline cells beyond GSCs and germline cysts

Both *sakura^null/null* and *sakura* germline RNAi knockdown driven by UAS-Dcr-2 and NGT-Gal4, which initiates RNAi in germline from GSCs, resulted in rudimentary ovaries (*Figures 2C and 5A*). These ovaries lacked later-stage germline cells and egg chambers, making it difficult to assess the role of Sakura beyond the germarium. However, because Sakura-EGFP expression is not limited to GSCs and germline cysts (*Figure 1E and F*), we speculated that Sakura might function in later-stage germline cells as well.

To explore this possibility, we used TOsk-Gal4 (a combination of osk-Gal4 and αTub67C-Gal4) to drive *sakura* RNAi knockdown in germline cells after the germline cyst stage (from germarium region 2b onward, *Figure 1A*), sparing GSCs and germline cysts (*ElMaghraby et al., 2022*). Unlike

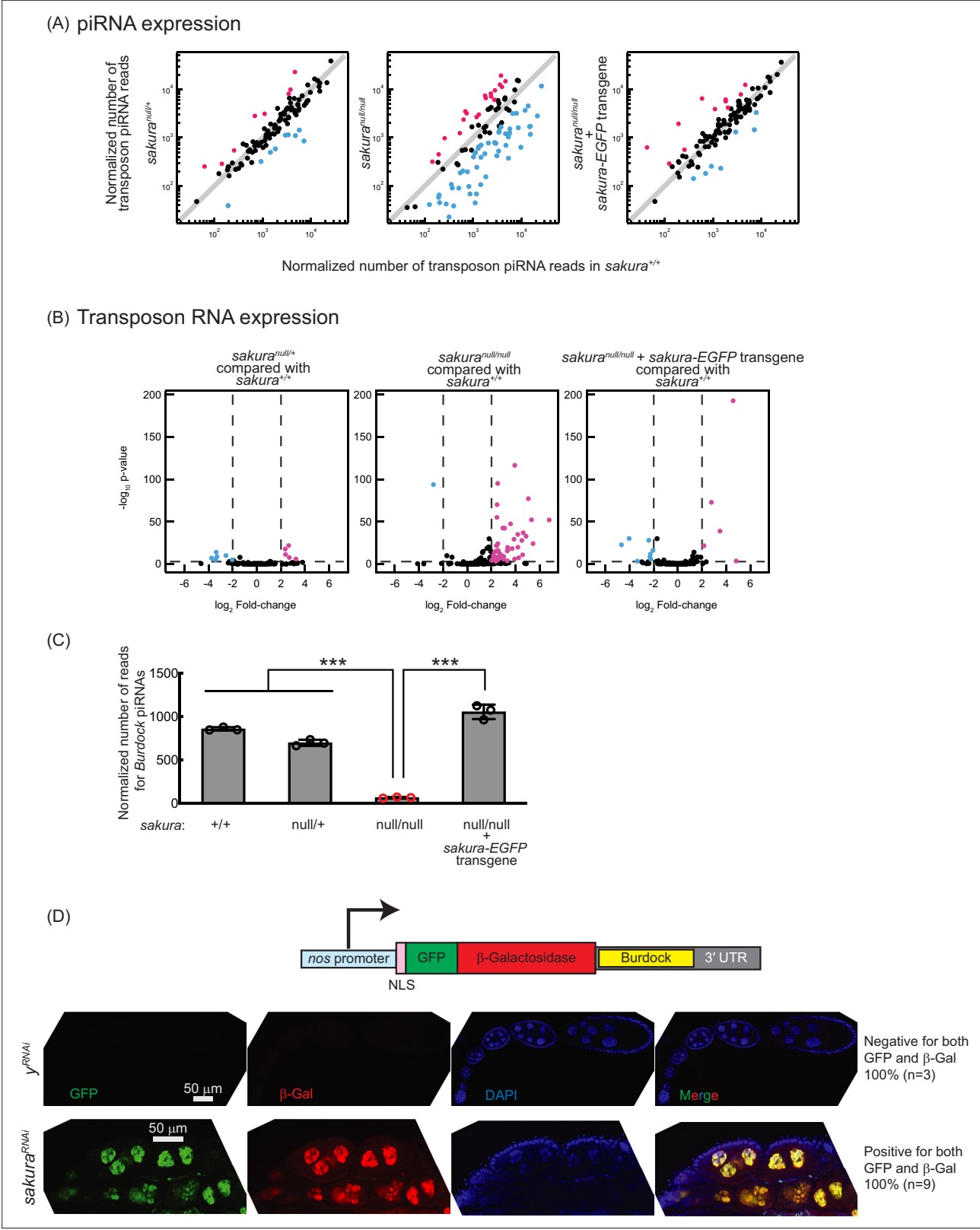

**Figure 4.** Loss of *sakura* results in lower piwi-interacting RNA (piRNA) levels and loss of piRNA-mediated transposon silencing. (**A**) Scatter plots of normalized number of transposon piRNA reads from small RNA-seq of indicated genotypes compared with *sakura^+/+^*. Means of three biological replicates are plotted. Downregulated (fold-change <0.5) and upregulated (fold-change >2) transposon piRNAs are shown in cyan and magenta, respectively. (**B**) Volcano plots of transposon RNAs from RNA-seq of indicated genotypes compared with *sakura^+/+^*. Three biological replicates per

*Figure 4 continued on next page*

*Figure 4 continued*

genotype were analyzed. Downregulated (adjusted p-value < 0.001 and log$_2$(fold-change) < –2) and upregulated (adjusted p-value < 0.001 and log$_2$(fold-change) >2) transposons are shown in cyan and magenta, respectively. (**C**) Normalized number of reads for *Burdock* piRNAs from small RNA-seq. Mean ± SD (n = 3). P-value < 0.001 (Student's t-test, unpaired, two-tailed) is indicated by ***. (**D**) The Burdock sensor harbors a nanos promoter, a nuclear localization signal (NLS) appended to GFP and β-gal coding sequences, and a target sequence for *Burdock* piRNAs in the 3'UTR. Confocal images of ovaries from control (*y$^{RNAi}$*) and *sakura$^{RNAi}$* flies harboring the Burdock sensor, where RNAi knockdown was specifically driven in the female germline with UAS-Dcr2 and NGT-Gal4. GFP (green), β-gal (red), and DAPI (blue). Scale bars: 50 μm. Three out of three tested control samples were negative for both GFP and β-gal, while 9 out of 9 tested *sakura$^{RNAi}$* samples were positive for both markers.

The online version of this article includes the following source data and figure supplement(s) for figure 4:

**Figure supplement 1.** Sex-specific alternative splicing of sex-lethal (*sxl*) is dysregulated in *sakura$^{null}$* ovaries.

**Figure supplement 1—source data 1.** Original uncropped gel image used in *Figure 4—figure supplement 1*, indicating the relevant bands.

**Figure supplement 1—source data 2.** Original uncropped, unedited gel image file used in *Figure 4—figure supplement 1*.

NGT-Gal4, TOsk-Gal4-driven *sakura* RNAi knockdown did not drastically affect ovary morphology (*Figure 5A*), allowing us to study the effects of *sakura* loss in egg chambers. We confirmed sakura RNAi (*sakura RNAi #1* and #2) knockdown efficiency by Western blot, showing effective depletion of Sakura in ovaries from both NGT-Gal4 and TOsk-Gal4-driven RNAi lines (*Figure 5B*).

Interestingly, *TOsk-Gal4*-driven *sakura* RNAi knockdown severely reduced the numbers of eggs laid (*Figure 5C*) and stage 14 oocytes in ovaries compared to control RNAi (*y$^{RNAi}$*) (*Figure 5D*), suggesting that *sakura* is important for oogenesis beyond the germline cyst stage. Additionally, we observed mislocalized and dispersed Oo18 RNA-binding protein (Orb), a marker for oocyte identity, in *TOsk-Gal4*-driven sakura RNAi ovaries beginning around stage 6 of oogenesis (*Figure 5E*).

In controls, Orb was enriched and properly localized to the posterior end of stage ~6–8 egg chambers (*Figure 5E*, yellow arrows). In *sakura$^{RNAi}$* ovaries, although Orb localization appeared normal at stage ~6 (yellow arrow), mislocalization was evident by stage ~8 (magenta arrow), and signs of cytoskeletal disorganization were apparent in later-stage egg chambers (white arrowheads). Phalloidin staining to visualize F-actin further supported cytoskeletal defects in *sakura$^{RNAi}$* later-stage egg chambers (*Figure 5—figure supplement 1*).

These defects likely contribute to the impaired production of stage 14 oocytes and eggs in *sakura$^{RNAi}$* flies (*Figure 5C and D*). We conclude that Sakura plays an essential role in oogenesis beyond the early germline stages. For all subsequent *sakura* RNAi experiments, we used the *sakura RNAi* #2 line.

## Sakura is required intrinsically for GSC establishment, maintenance, and differentiation

To investigate whether *sakura* functions autonomously in the germline, we performed mosaic analysis using the FLP-FRT system with heat shock promoter-driven FLP (hs-flp) (*Xu and Rubin, 1993*; *Rubin and Huynh, 2015*). First, to assess its role in GSC establishment, we induced *sakura$^{null}$* clones in primordial germ cells (PGCs) by applying heat shock before the early third instar larval stage and tracked their development into adult GSCs, following a previously established protocol (*Yang et al., 2007*). In control adult flies (*FRT82B*), 12.3% (20/163) of GSCs were marked, whereas only 1.9% (4/213) were marked in *sakura$^{null}$* mutants (*FRT82B, sakura$^{null}$*) (p-value = 4.8 * 10$^{-11}$, chi-square test), indicating that *sakura* is autonomously required for GSC establishment.

Next, we assessed GSC maintenance using the same system by generating *sakura$^{null}$* GSC clones in adult flies and measuring clone loss over time, as described previously (*Xie and Spradling, 1998*; *Yang et al., 2007*). Four days after clone induction, 27.8% of GSCs were marked in controls (*FRT82B*) and 22.1% in *sakura$^{null}$* (*FRT82B, sakura$^{null}$*), which we defined as initial levels (*Figure 6A*). In controls, the percentage of marked GSCs declined to 25.8% on day 7 and 16.8% on day 14, resulting in a 39.4% loss rate over 10 days (from day 4 to day 14). In contrast, the proportion of marked *sakura$^{null}$* GSC clones dropped more sharply—to 16.3% on day 7 and 5.4% on day 14—yielding a higher loss rate of 75.6% over 10 days. These results demonstrate that *sakura* is intrinsically important for GSC maintenance.

We further tested whether *sakura* is intrinsically required for GSC differentiation. We quantified the number of marked (GFP-negative) and unmarked (GFP-positive) GSC-like cells (germline cells with a round spectrosome) in the germaria containing marked GSC clones at days 4, 7, and 14 post-induction

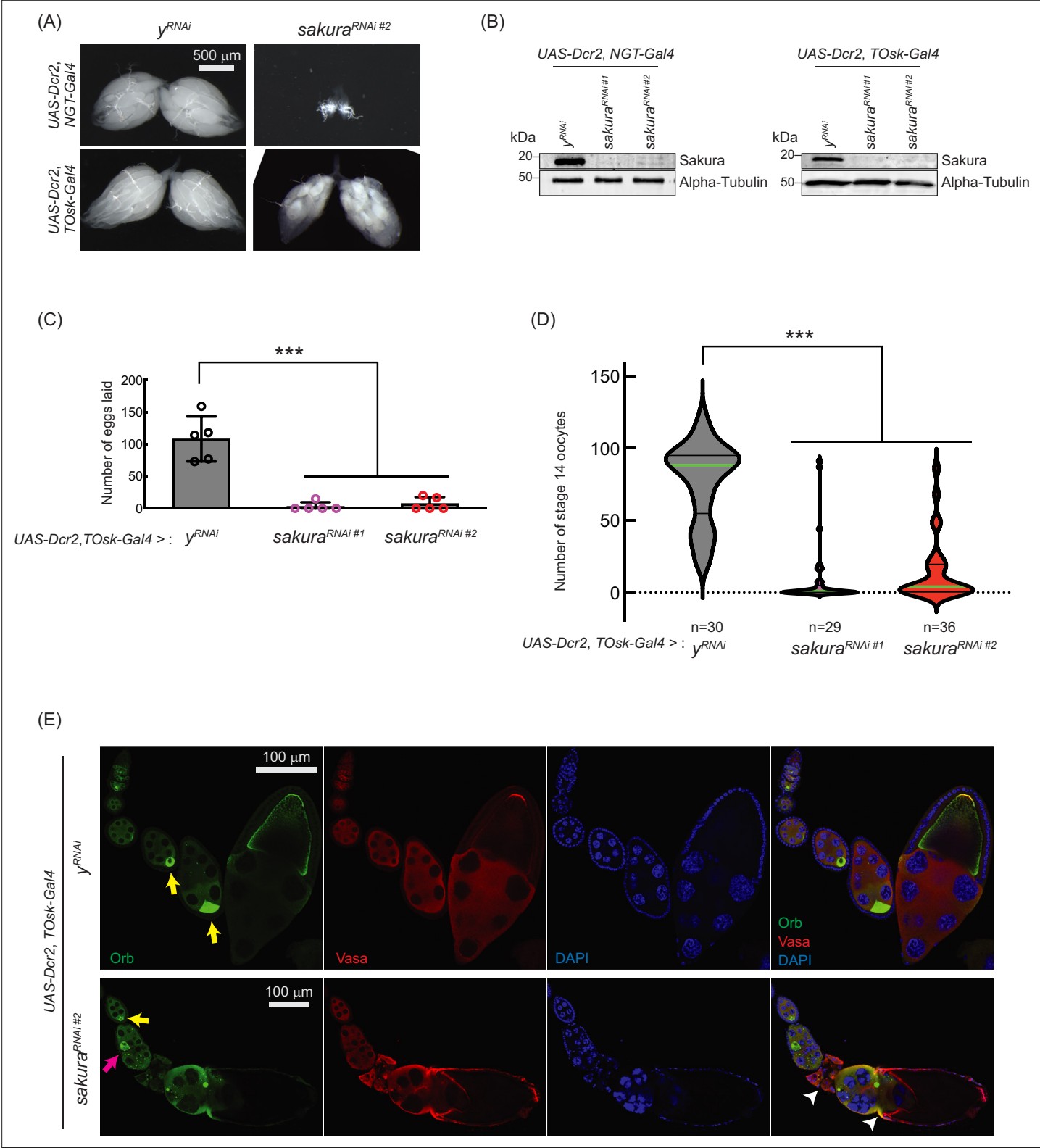

**Figure 5.** *sakura* is important for oogenesis in later-stage egg chambers. (**A**) Stereomicroscope images of dissected whole ovaries. Scale bar: 500 µm. (**B**) Western blot of dissected ovary lysates. *sakura*^RNAi #1^ and *sakura*^RNAi #2^ are two independent RNAi lines. (**C**) Number of eggs laid by *sakura* RNAi knockdown driven by UAS-Dcr2 and TOsk-Gal4. Mean ± SD (n = 5). P-value < 0.001 (Student's t-test, unpaired, two-tailed) is indicated by ***. (**D**) Number of stage 14 oocytes in ovaries of *sakura* RNAi knockdown flies driven by *UAS-Dcr2* and *TOsk-Gal4*. The numbers of stage 14 oocytes per fly (per a pair of ovaries) are shown. n is the number of flies examined. P-value < 0.001 (Student's t-test, unpaired, two-tailed) is indicated by ***.

*Figure 5 continued on next page*

*Figure 5 continued*

(**E**) Confocal images of the ovaries from *sakura* RNAi knockdown flies driven by *UAS-Dcr2* and *TOsk-Gal4*, stained with anti-Oo18 RNA-binding protein (Orb) and anti-Vasa antibodies. Orb (green), Vasa (red), and DAPI (blue). Yellow arrows label the normal enrichment of Orb in the developing oocytes. Magenta arrow labels mislocalized developing oocyte. White arrowheads label egg chambers with cytoskeletal disorganization. Scale bars: 100 μm. In A-E, *y-RNAi* was used as a control.

The online version of this article includes the following source data and figure supplement(s) for figure 5:

**Source data 1.** Original uncropped gel blot images used in *Figure 5B*, indicating the relevant bands.

**Source data 2.** Original uncropped, unedited gel blot image files used in *Figure 5B and D*.

**Figure supplement 1.** *sakura* is important for oogenesis in later-stage egg chambers.

---

(*Figure 6B*). We found that marked *sakura^{null}* GSC-like cells were significantly more numerous than marked control GSC-like cells at all time points (*Figure 6C*). The number of marked *sakura^{null}* GSC-like cells increased over the 10 day period, while the number of marked control cells did not (*Figure 6C*). In contrast, unmarked GSC-like cells in germaria containing *sakura^{null}* GSC clones did not differ significantly in number compared with those in germaria containing marked control GSC clones, and the number did not increase over time (*Figure 6—figure supplement 1*).

These results demonstrate that *sakura* is intrinsically important in germline cells, including GSCs, to regulate proper division and differentiation. In the absence of intrinsic *sakura*, germline cells become tumorous and undergo uncontrolled proliferation. Taken together, we conclude that *sakura* is required intrinsically for GSC establishment, maintenance, and differentiation.

## Loss of *sakura* inhibits Dpp/BMP signaling

The Dpp/BMP signaling pathway plays a central role in regulating *bam* expression and GSC self-renewal and differentiation (*Figure 7A*; *Kirilly and Xie, 2007*; *Hayashi et al., 2020*). We hypothesized that the germless and tumorous phenotypes observed in *sakura* loss-of-function ovaries could be due to Dpp/BMP signaling misregulation. To test this, we knocked down *sakura* in the germline driven by *UAS-Dcr2* and *NGT-Gal4* in the presence of *bam-GFP* reporter (*Chen and McKearin, 2003b*). In controls (y^{RNAi}), Bam-GFP expression was restricted to 8-cell cysts and disappeared in 16 cell cysts and onward (*Figure 7B*). In contrast, sakura knockdowns ovaries showed persistent Bam-GFP expression throughout the germarium, including the GSC niche region (*Figure 7B*).

To further confirm this finding, we used the FLP-FRT system to induce *sakura^{null}* clones in germaria and performed immunostaining with anti-Bam antibody. In control clones (*FRT82B*), Bam expression was observed exclusively in 8 cell cysts. However, in all *sakura^{null}* clones (*FRT82B, sakura^{null}*), Bam was aberrantly expressed throughout the germarium, including in GSCs (*Figure 7C*). These results suggest that in the absence of *sakura*, Bam expression is no longer repressed by Dpp/BMP signaling in the GSC niche, leading to GSC loss, and is no longer shut off after the 16 cell cyst stage.

In GSCs, pMad translocates into the nucleus to repress Bam expression (*Figure 7A*; *Kai and Spradling, 2003*; *Song et al., 2004*). We stained ovaries with anti-pMad antibodies and found that pMad intensity was significantly reduced in *sakura^{null}* GSC clones compared to control GSCs (*Figure 7D and E*). This indicates that the dysregulated Bam expression observed in *sakura^{null}* was due to reduced pMad levels in GSCs.

Previous studies have shown that ectopic expression of a stable form of CycA leads to germ cell loss (*Chen et al., 2009*). This germ cell loss phenotype is also observed upon ectopic *bam* expression in GSCs (*Xie and Spradling, 1998*; *Chen and McKearin, 2003a*; *Xia et al., 2010*). It was reported that Bam associates with Otu to promote deubiquitination and stabilization of CycA (*Ji et al., 2017*). Since we observed derepressed *bam* expression in *sakura^{null}* cells, we investigated whether CycA levels were affected. Staining ovaries with anti-CycA antibodies revealed higher CycA intensity in *sakura^{null}* clones compared to neighboring wild-type cells in the same germarium (*FRT82B, sakura^{null}*) and control clones in control germarium (*FRT82B*) (*Figure 7F*). *sakura^{null}* clones exhibited significantly higher mean CycA intensity compared to control clones (*Figure 7G*). The elevated CycA levels in *sakura^{null}* clones likely result from Bam misexpression and Bam-mediated stabilization.

To determine whether *bam* misexpression occurs in egg chambers in *sakura* knockdown conditions, we examined *bam* expression in TOsk-Gal4-driven *sakura* RNAi flies. Bam was appropriately restricted to 8 cell cysts in germaria and not misexpressed in egg chambers compared to yRNAi

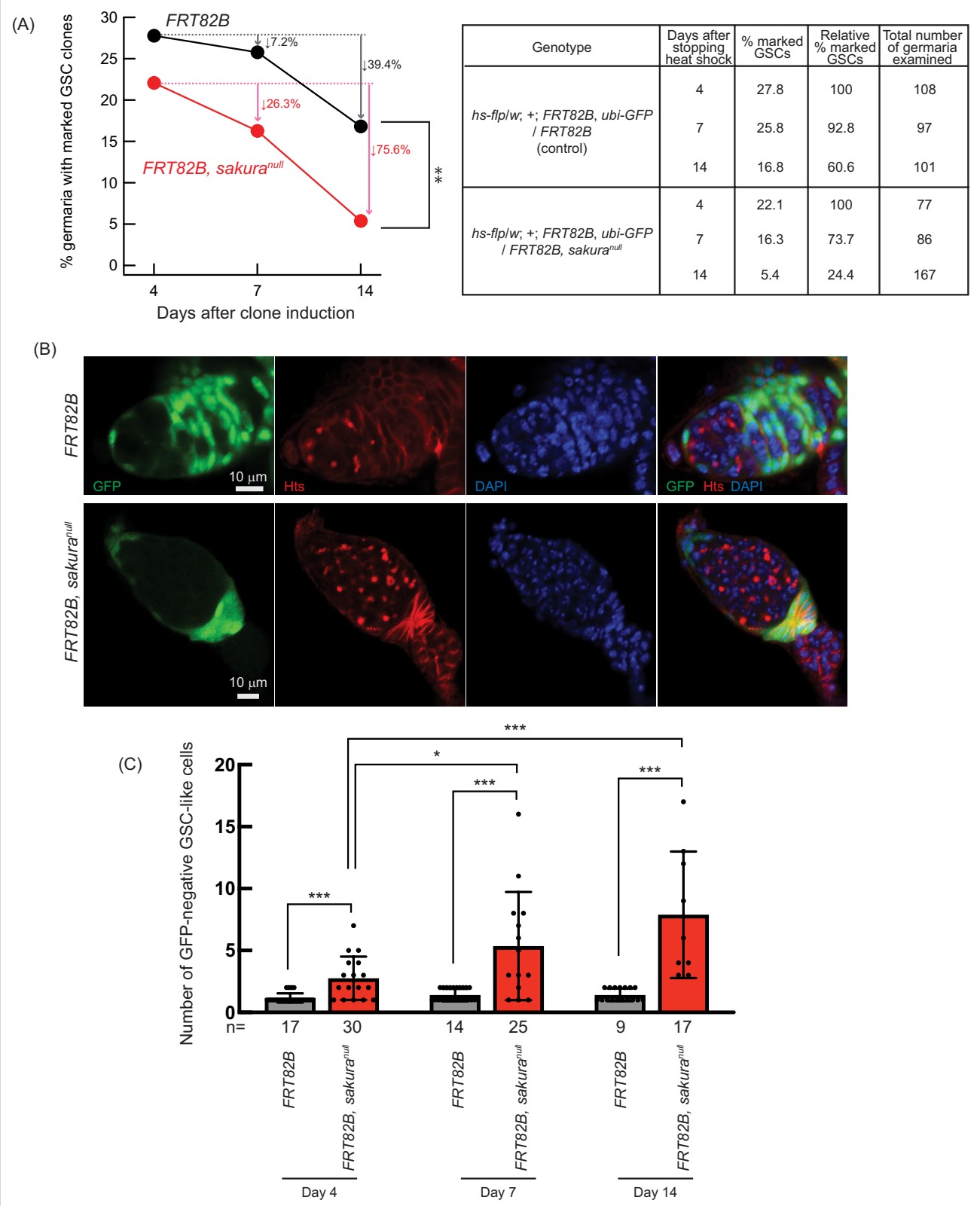

**Figure 6.** Germline clonal analysis of *sakura^null^*. (**A**) Percentage of germaria with marked germline stem cell (GSC) clones indicated by the absence of GFP at 4, 7, and 14 days after clone induction at the adult stage. Arrows indicate the percent decrease of marked GSC clones compared to day 4. The genotype, actual percentage, and total number of germaria examined are shown in the adjacent table. P-value < 0.01 (Chi-squared test) is indicated by **. (**B**) Confocal images of germ cell clones. *sakura^null^* and control clones were marked by the absence of GFP. GFP (green), Hts (red), and DAPI (blue).

*Figure 6 continued on next page*

*Figure 6 continued*

Scale bars: 10 µm. (**C**) Number of marked (GFP-negative) GSC-like cells in germaria with marked GSCs of the indicated genotypes at 4, 7, and 14 days after clone induction. Mean ± SD. GSC-like cells containing round spectrosomes were identified through immunostaining with anti-Hts antibody. P-value < 0.05 and < 0.001 (Student's t-test, unpaired, two-tailed) is indicated by * and ***, respectively.

The online version of this article includes the following figure supplement(s) for figure 6:

**Figure supplement 1.** *sakura^null^* clone germline cells intrinsically cause tumorous phenotype.

controls (*Figure 7—figure supplement 1*, cyan arrowheads). Thus, *sakura* is specifically required in GSCs and cyst cells to regulate *bam* expression.

We also examined DE-Cadherin (DE-Cad), which plays a role in oocyte positioning within the egg chamber (Godt and Tepass 1998). DE-Cad level and localization appeared normal in both control and *sakura* RNAi ovaries through approximately stage 8. However, later-stage egg chambers in sakura RNAi flies displayed structural disorganization (*Figure 7—figure supplement 1*, white arrowheads), consistent with defects observed in *Figure 5E*.

## Attempts to rescue *sakura* loss-of-function ovariole phenotypes

To determine whether the phenotypes caused by *sakura* loss-of-function could be rescued, we first performed genetic interaction experiments by knocking down either *bam* or cycA in the germline. As expected, germline-specific *bam* RNAi driven by UAS-Dcr2 and NGT-Gal4 resulted in 100% tumorous ovarioles, while cycA RNAi produced 100% normal ovarioles in 2–5 day-old flies (*Figure 7—figure supplement 2*). Double knockdown of *sakura* and *bam* reduced the proportion of germless ovarioles and increased the proportion of tumorous ovarioles compared to sakura RNAi alone or the double knockdown of sakura and control w RNAi, while no normal ovarioles were observed (*Figure 7—figure supplement 2*). *sakura* and cycA double knockdown had no effect on the phenotype distribution, suggesting that the germless phenotype in *sakura* mutants is partially attributable to ectopic *bam* expression but not to cycA misregulation.

The number of GSC-like cells was increased in *sakura* and *bam* double knockdowns compared to *sakura* RNAi or *sakura* and w double RNAi, whereas sakura and cycA knockdown did not alter GSC-like cell numbers (*Figure 7—figure supplement 2*), further supporting a genetic interaction between *sakura* and *bam*.

Next, we tested whether the *sakura* loss-of-function phenotypes could be rescued by overexpressing components of the Dpp/BMP signaling pathway. Germline expression of UASp-Mad-GFP driven by nos-Gal4-VP16 in both *sakura^null/+^* controls and *sakura^null/null^* did not alter the proportions of ovariole phenotypes or the number of GSC-like cells (*Figure 7—figure supplement 3*). Thus, transgenic overexpression of Mad did not rescue the *sakura* mutant phenotypes.

We then tested whether overexpression of a constitutively active form of the Dpp receptor Thickveins (Tkv.Q253D) (*Casanueva and Ferguson, 2004*) could rescue the phenotypes. Expression of *UASp-tkv.Q253D* in the germline using an *NGT-Gal4* in a control (*y^RNAi^*) background resulted in 100% tumorous ovarioles, as expected (*Figure 7—figure supplement 4*). When co-expressed with *sakura* RNAi, Tkv.Q253D increased the proportion of tumorous ovarioles and decreased the proportion of germless ovarioles relative to *sakura^RNAi^* alone but did not restore normal ovarioles. The number of GSC-like cells was also increased by Tkv.Q253D expression in *sakura^RNAi^* (*Figure 7—figure supplement 4*),indicating partial phenotypic modulation via BMP pathway activation.

## Sakura binds Otu

To explore the molecular function of Sakura in oogenesis, we aimed to identify Sakura-interacting proteins. We conducted a co-immunoprecipitation experiment using anti-GFP magnetic beads on the ovary lysates of flies expressing Sakura-EGFP, followed by mass spectrometry to identify co-immunoprecipitated proteins. Ovary lysates from flies without the *sakura*-EGFP transgene (w1118) served as negative controls. Mass spectrometric analysis revealed several proteins with peptide signals present in all three biological replicates of the Sakura-EGFP samples without any signals in any of the three biological replicates of the negative control (*Table 1*). Among these, the most promising candidate, exhibiting the highest unique peptide signal in each of the three replicates of the Sakura-EGFP samples was Otu.

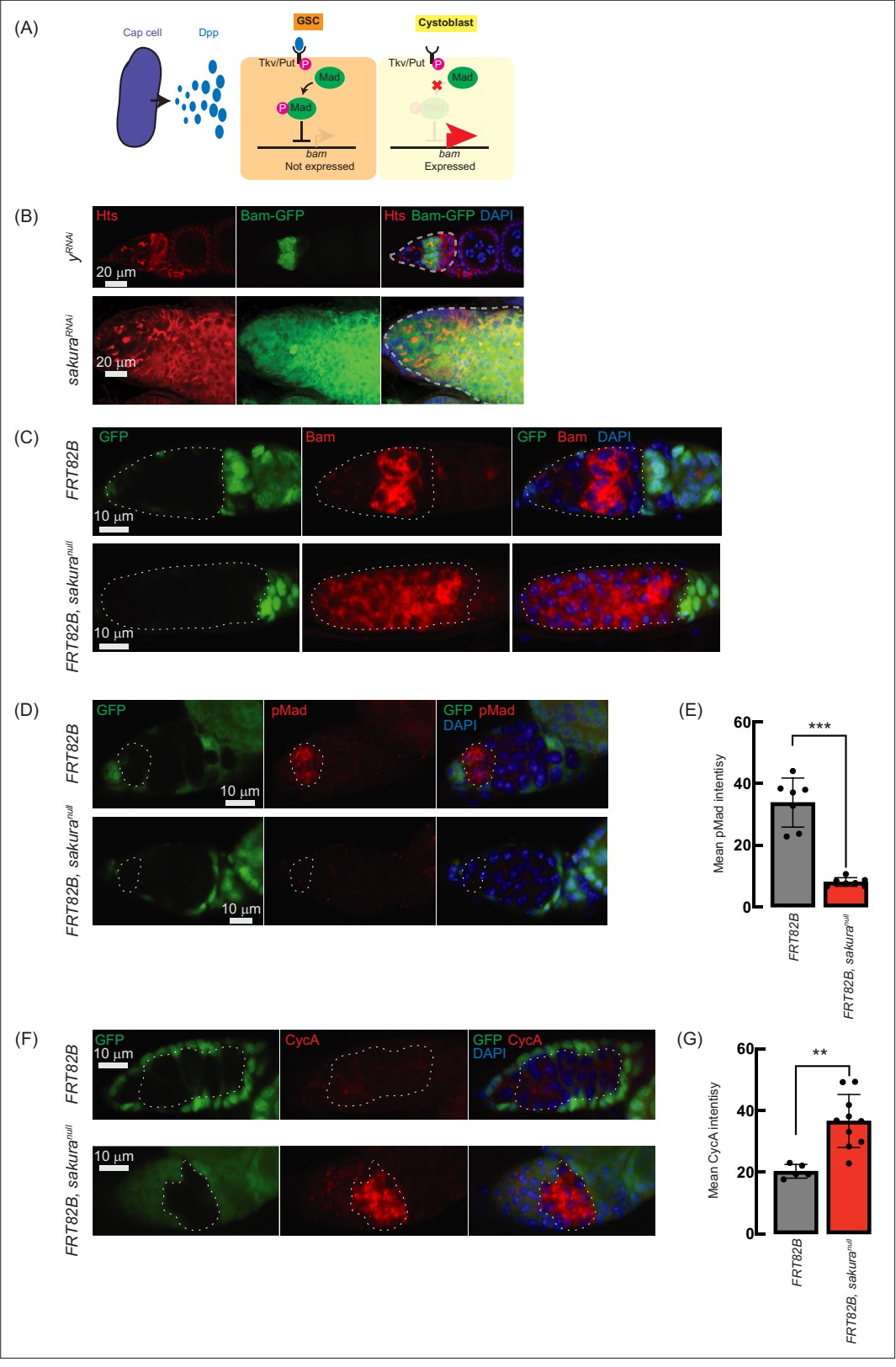

**Figure 7.** Loss of *sakura* inhibits Dpp/bone morphogenetic protein (BMP) signaling. (**A**) Schematic illustration of *bam* expression regulation. Cap cells secrete diffusible Decapentaplegic (Dpp), which is received by its receptor, a heterodimer of Thick vein (Tkv) and Punt (Put), in germline stem cells (GSCs). The activated Dpp signaling eventually phosphorylates Mother-against-dpp (Mad). The phosphorylated Mad (pMad) represses the transcription

*Figure 7 continued on next page*

*Figure 7 continued*

of bag-of-marbles (bam). The repression of *bam* in GSCs is crucial for maintaining their stemness. Cystoblasts do not receive Dpp, and Bam expression is crucial for promoting cystoblast differentiation from GSCs. (**B**) Confocal images of ovaries from control (*y^RNAi*) and *sakura^RNAi* flies harboring the *bam-GFP* reporter, where RNAi knockdown was specifically driven in the female germline with *UAS-Dcr2* and *NGT-Gal4*. Bam-GFP (green), Hts (red), and DAPI (blue). Germarium are outline ad by dotted line. Scale bars: 20 µm. (**C, D**) Confocal images of germaria with germline clones of sakura^null^ stained with (**C**) anti-Bam antibody and (**D**) anti-pMad antibody. GFP (green), Bam or pMad (red), and DAPI (blue). Scale bars: 10 µm. (**E**) Mean pMad intensity in the germline clones of the indicated genotypes. Mean ± SD (n = 7). P-value < 0.001 (Student's t-test, unpaired, two-tailed) is indicated by *** (**F**) Confocal images of germaria with germline clones of *sakura^null* stained with anti-CycA antibody. GFP (green), CycA (red), and DAPI (blue). Scale bars: 10 µm. (**G**) Mean CycA intensity in the germline clones of the indicated genotypes. Mean ± SD (n=5 and 10 for *FRT82B* and *FRT82B, sakura^null*, respectively). P-value < 0.01 (Student's t-test, unpaired, two-tailed) is indicated by **. In C, D, and F, clones were marked with the absence of GFP.

The online version of this article includes the following figure supplement(s) for figure 7:

**Figure supplement 1.** Bag-of-marbles (Bam) is not misexpressed in TOsk-Gal4 >*sakura^RNAi* egg chambers.

**Figure supplement 2.** Ratio of germless and tumorous phenotypes from double RNAi knockdown of *sakura* with bag-of-marbles (*bam), cyclin A (cycA),* or *ovarian tumor (otu)*.

**Figure supplement 3.** nos-Gal4-VP16>UASp-Mad-GFP did not rescue the *sakura^null* phenotypes.

**Figure supplement 4.** Ratio of germless and tumorous phenotypes in *sakura RNAi* knockdown and *NGT-Gal4 >UASp tkv*.

To confirm this finding, we generated a polyclonal anti-Otu antibody and validated it by Western blots using ovary lysates from the hypomorphic *otu^14* mutant (*Figure 8—figure supplement 1*; *King et al., 1986*; *Storto and King, 1987*; *Steinhauer and Kalfayan, 1992*; *Pauli et al., 1993*). Using this antibody, we detected endogenous Otu protein in the Sakura-EGFP immunoprecipitants from ovaries of 3–7 day-old flies, but not in the negative control (*Figure 8A*), validating the mass spectrometry results. Next, using wild-type (without Sakura-EGFP) ovary lysates from 3–7 day-old-flies, we immunoprecipitated endogenous Sakura with anti-Sakura antibody and detected endogenous Otu in the immunoprecipitant by Western blot (*Figure 8B*), demonstrating that endogenous Sakura interacts with endogenous Otu. Furthermore, we found that ectopically expressed Flag-tagged Otu co-immunoprecipitated with ectopically expressed HA-tagged Sakura in S2 cells (*Figure 8C*).

**Table 1. Number of unique peptide counts detected by mass spec**.
Proteins with peptide signals present in all three biological replicates of the Sakura-EGFP samples without any signals in any of the three biological replicates of the negative control are shown.

| Identified protein name | Gene ID | Unique peptide counts | | | | | |
| --- | --- | --- | --- | --- | --- | --- | --- |
| | | Sakura-EGFP samples | | | w1118 samples (negative control) | | |
| | | Replicate 1 | Replicate 2 | Replicate 3 | Replicate 1 | Replicate 2 | Replicate 3 |
| Ovarian tumor (Otu) | CG12743 | 44 | 38 | 18 | 0 | 0 | 0 |
| Uncharacterized protein Dmel_CG4679 | CG4679 | 17 | 8 | 2 | 0 | 0 | 0 |
| Uncharacterized protein Dmel_CG14997 | CG14997 | 17 | 7 | 2 | 0 | 0 | 0 |
| Mitochondrial ribosomal protein S22 | CG12261 | 14 | 6 | 1 | 0 | 0 | 0 |
| Tudor | CG9450 | 15 | 1 | 1 | 0 | 0 | 0 |
| Mitochondrial ribosomal protein S5 | CG40049 | 10 | 4 | 1 | 0 | 0 | 0 |
| HECT and RLD domain containing protein 2 | CG11734 | 4 | 9 | 1 | 0 | 0 | 0 |
| Mitochondrial ribosomal protein S10 | CG4247 | 5 | 3 | 1 | 0 | 0 | 0 |
| Mitochondrial ribosomal protein S31 | CG5904 | 3 | 1 | 1 | 0 | 0 | 0 |
| Uncharacterized protein Dmel_CG1316 | CG1316 | 2 | 1 | 1 | 0 | 0 | 0 |

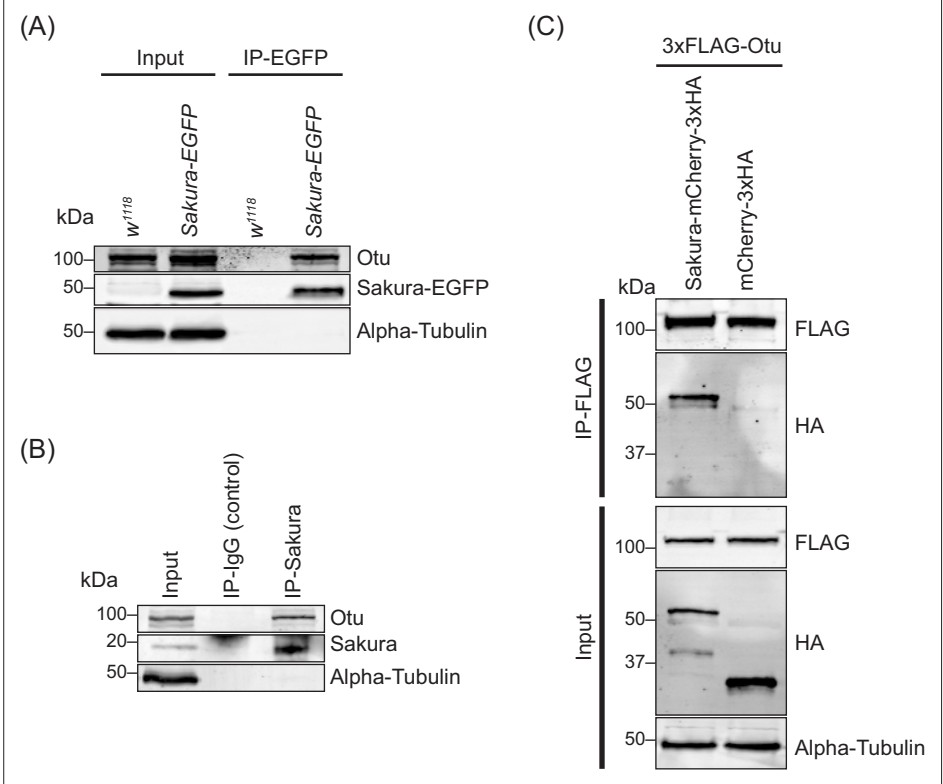

**Figure 8.** Sakura interacts with ovarian tumor (Otu). (**A**) Co-immunoprecipitation using anti-GFP magnetic beads followed by Western blotting. Ovary lysates expressing *Sakura-EGFP* in *sakura⁺/⁺* background and those from *w¹¹¹⁸* negative control were tested. (**B**) Co-immunoprecipitation using beads bound with rabbit anti-Sakura followed by Western blotting. Ovary lysates from *w¹¹¹⁸* flies were used. Rabbit IgG was used as a control IP. (**C**) Co-immunoprecipitation using beads bound with anti-FLAG antibody followed by Western blotting. S2 cell lysates expressing 3xFLAG-Otu and Sakura-mCherry-3xHA or mCherry-3xHA (negative control) were used.

The online version of this article includes the following source data and figure supplement(s) for figure 8:

**Source data 1.** Original uncropped gel blot images used in *Figure 8A, B, C* and *Figure 8—figure supplement 1* indicating the relevant bands.

**Source data 2.** Original uncropped, unedited gel blot image files used in *Figure 8A, B, C* and *Figure 8—figure supplement 1*.

**Figure supplement 1.** Anti-ovarian tumor (Otu) western blot of dissected ovary lysates.

---

The *otu* gene encodes two annotated protein isoforms, a predominant 98 kDa isoform and a less abundant 104 kDa isoform generated by alternative splicing (*Steinhauer and Kalfayan, 1992*). The 98 kDa isoform lacks the Tudor domain, which is encoded by an alternatively spliced 126-nucleotide exon (*Figure 9B*. Otu(ΔTudor)) (*van Buskirk and Schüpbach, 2002*). Previous studies have shown that the 104 kDa Otu isoform is more abundant in predifferentiated germline cells and is sufficient to carry all known Otu functions, whereas the 98 kDa Otu isoform becomes prominent during later stages of oogenesis and is implicated in nurse cell regression and oocyte maturation (*Sass et al., 1995*). Consistent with these findings, we observed that ovaries from very young (2–5 hr old) flies—containing only germaria and previtellogenic egg chambers—predominantly expressed the 104 kDa isoform (*Figure 8—figure supplement 1*). In contrast, ovaries from 3–7 day-old flies—containing all 14 stages of oogenesis—predominantly expressed the 98 kDa isoform. Importantly, both the 104 kDa and 98 kDa isoforms co-immunoprecipitated with endogenous Sakura using anti-Sakura antibody, demonstrating that Sakura interacts with both isoforms of Otu and that the Tudor domain is not required for this interaction (*Figure 8—figure supplement 1*).

Next, we sought to identify which regions of Sakura and Otu are important for their interaction. We predicted the structure of the Sakura-Otu protein complex using AlphaFold (*Jumper et al., 2021*; *Yang et al., 2023*). AlphaFold suggested that the N-terminal region of Sakura (1-49aa) and the

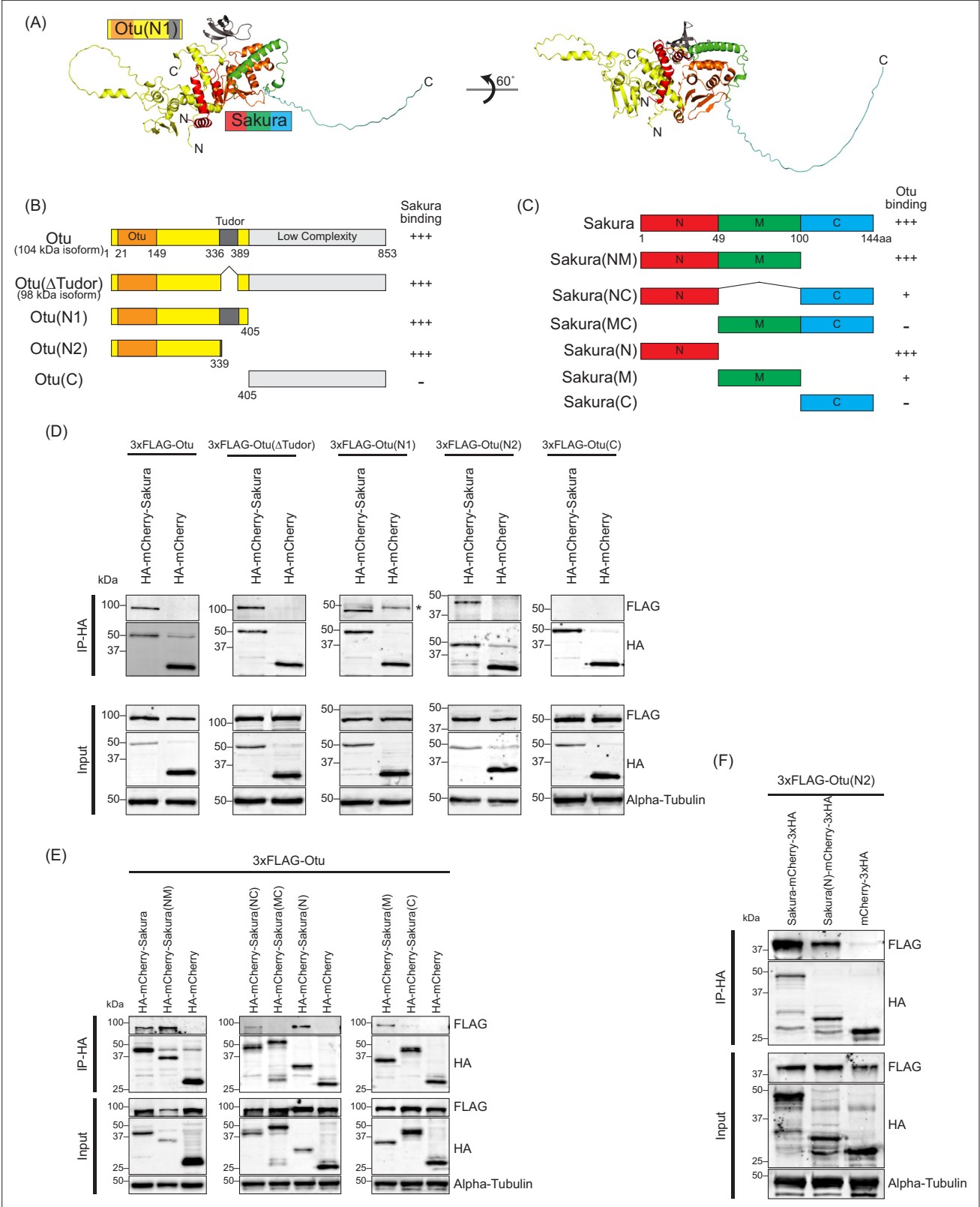

**Figure 9.** N-terminal regions of Sakura and ovarian tumor (Otu) are important for interaction. (**A**) Predicted structure of the Sakura and Otu protein complex made by AlphaFold. Full-length Sakura and N-terminal Otu fragment (N1. 1-405aa) were used for prediction. (**B**) Full-length Otu and Otu fragments tested in co-immunoprecipitation assays. The binding assay results from (**D**) is summarized. (**C**) Full-length Sakura and Sakura fragments tested in co-immunoprecipitation assays (N: N-terminal, M: middle, C: C-terminal). The binding assay results from (**E**) is summarized. (**D, E, F**) Co-

*Figure 9 continued on next page*

*Figure 9 continued*

immunoprecipitation using anti-HA magnetic beads followed by Western blotting. S2 cell lysates expressing HA-tagged mCherry were used as negative controls.

The online version of this article includes the following source data and figure supplement(s) for figure 9:

**Source data 1.** Original uncropped gel blot images used in *Figure 9D, E, F*, *Figure 9—figure supplement 1* indicating the relevant bands.

**Source data 2.** Original uncropped, unedited gel blot image files used in *Figure 9D, E, F*, *Figure 9—figure supplement 1*.

**Figure supplement 1.** Co-immunoprecipitation assay to test interaction between N-terminal fragments of Sakura and ovarian tumor (Otu) in S2 cells.

N-terminal region of Otu (1-405aa) are highly likely to form interactions, while the C-terminal region of Sakura (100-144aa) does not interact with Otu (*Figure 9A–C*). The Tudor domain of Otu is not directly involved in this interaction, consistent with our findings (*Figure 8—figure supplement 1*). To experimentally validate the predictions made by AlphaFold, we generated a series of truncated Otu fragments, including Otu(ΔTudor) (1–336, 389-853aa, corresponding to the endogenous 98 kDa isoform), Otu(N1) (1-405aa), Otu(N2) (1-339aa), and Otu(C) (405-853aa) (*Figure 9B*). We then performed co-immunoprecipitation assays to test which fragments could interact with full-length Sakura. All fragments except Otu(C) were associated with full-length Sakura protein in S2 cells (*Figure 9D*), demonstrating that the C-terminal low-complexity region of Otu is dispensable for the interaction. This assay also showed that the Tudor domain is not critical and the N-terminal fragment (1-339aa) of Otu is sufficient for interaction with Sakura. These findings align well with the predictions made by AlphaFold.

Subsequently, we created several truncated Sakura fragments and conducted co-immunoprecipitation assays in S2 cells to determine which Sakura fragments can interact with full-length Otu. The fragments generated included Sakura(NM) (1-100aa), Sakura(NC) (1–49, 100-144aa), Sakura(MC) (49-144aa), Sakura(N) (1-49aa), Sakura(M) (49-100aa), and Sakura(C) (100-144aa) (*Figure 9C*). We found that Sakura(NM), Sakura(NC), Sakura(N), and Sakura(M) were associated with Otu (*Figure 9E*). Compared to Sakura(NM) and Sakura(N), the interaction of Sakura(M) with Otu is relatively weaker, as indicated by a fainter band in the Western blot (*Figure 9E*). Sakura-NC exhibited a weak interaction with Otu, while Sakura(MC) and Sakura(C) showed no interaction (*Figure 9E*). These assays demonstrated that the N-terminal region (1-49aa) of Sakura is sufficient for interaction with Otu. Moreover, the results suggest that the C-terminal region of Sakura (100-144aa) does not interact with Otu and that adjoining it with the N-terminal (1-49aa) or the middle (M) region (49-100aa) weakens the interaction with Otu. These findings support the AlphaFold predictions.

Finally, to test whether the interaction between Sakura and Otu can be achieved solely through their N-terminal regions, we performed a co-immunoprecipitation assay with Sakura (N) and Otu (N2). We discovered that Sakura (N) and Otu (N2) physically interact (*Figure 9F*). A reciprocal co-immunoprecipitation assay also confirmed their interaction (*Figure 9—figure supplement 1*), revealing that the interaction between Sakura and Otu can be established with just the 1-49aa of Sakura and the 1-339aa of Otu. Therefore, we conclude that Sakura interacts with Otu in vivo, and the N-terminal regions of both proteins are sufficient for their interaction.

## Loss of *otu* phenocopies loss of *sakura*

Having established the interaction between Sakura and Otu, we were interested in whether they function together during oogenesis. We speculated that for them to interact and function together, Sakura and Otu must exhibit similar expression and localization patterns in ovaries. We generated two transgenic flies carrying full-length *otu-EGFP* or *otu(ΔTudor)-EGFP* transgenes; both constructs are C-terminally fused with EGFP and under the control of the *otu* promoter. We found that both Otu-EGFP and Otu(ΔTudor)-EGFP display similar expression and localization patterns to Sakura-EGFP (*Figures 1E, F and 10*). Like Sakura-EGFP, Otu-EGFP and Otu(ΔTudor)-EGFP are specifically expressed in germ cells, but not in follicle cells, and are localized to the cytoplasm, being enriched in the developing oocytes (*Figure 10A*). We did not observe any difference in localization patterns between Otu-EGFP and Otu(ΔTudor)-EGFP; suggesting that the Tudor domain is not crucial for Otu protein localization. Notably, the localization pattern exhibited by the transgenic Otu-EGFP generated in this study is consistent with a previous report (*Glenn and Searles, 2001*).

Previous studies have shown that mutations in *otu* lead to defects in germ cell division and differentiation, resulting in phenotypes including tumorous ovarioles (*King and Riley, 1982*; *King et al.,*

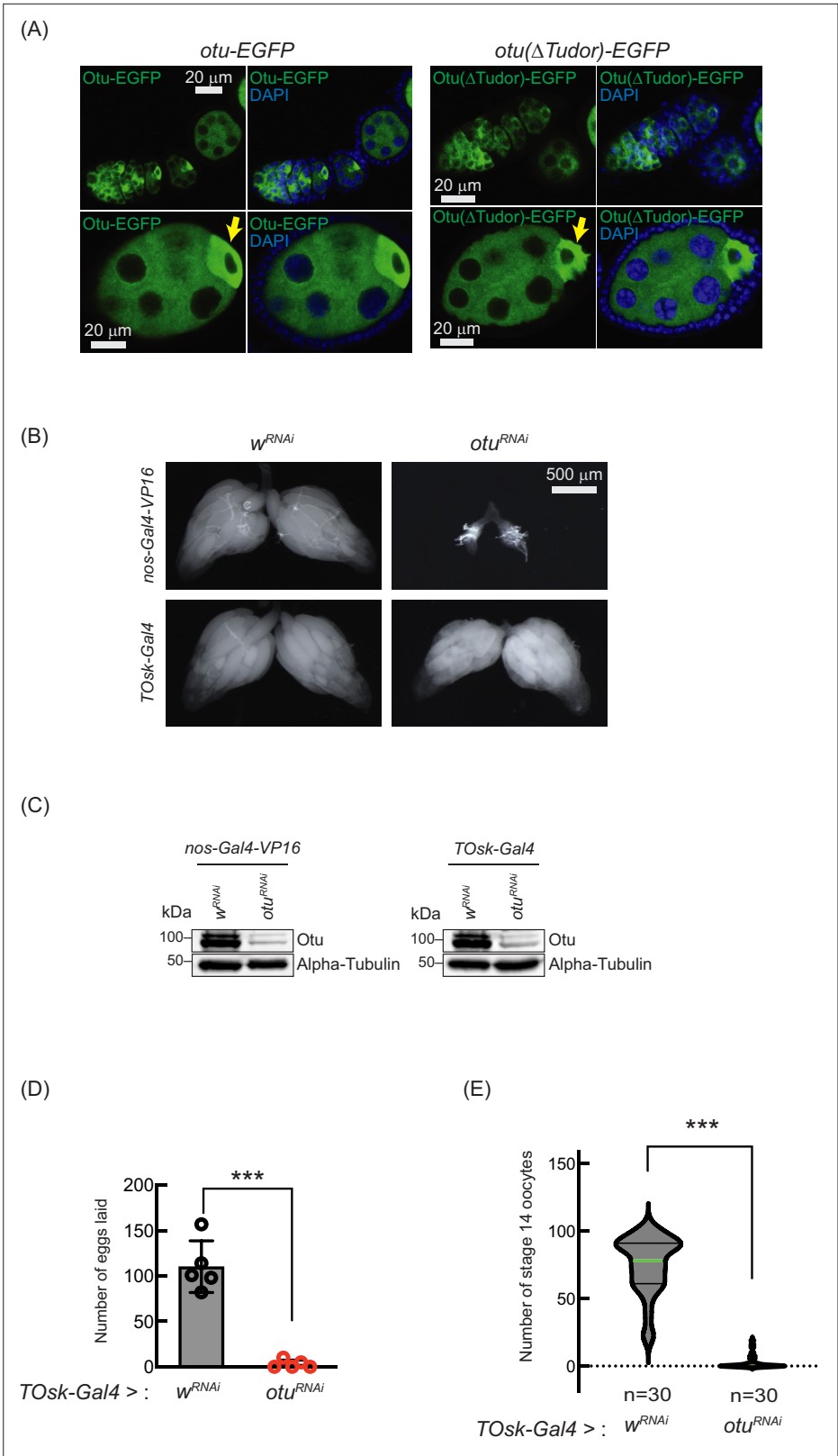

**Figure 10.** Loss of ovarian tumor (*otu)* phenocopies loss of *sakura*. (**A**) Confocal images of ovaries from *otu-EGFP* and *otu(ΔTudor)-EGFP* transgenic flies. Otu-EGFP and Otu(ΔTudor)-EGFP (green), DAPI (blue). Yellow arrows show the enrichment of Otu-EGFP and Otu(ΔTudor)-EGFP in developing oocytes. Scale bars: 20 µm. (**B**) Stereomicroscope images of dissected whole ovaries from *w^RNAi^* (control) and *otu^RNAi^* flies where RNAi

*Figure 10 continued on next page*

*Figure 10 continued*

knockdown was driven in the female germline with nos-Gal4-VP16 or *TOsk-Gal4*. Scale bar: 500 μm. (**C**) Western blot of dissected ovary lysates. (**D**) Number of eggs laid by *TOsk-Gal4 >w^RNAi* and *TOsk-Gal4 >otu^RNAi* flies. Mean ± SD (n = 5). P-value < 0.001 (Student's t-test, unpaired, two-tailed) are indicated by ***. (**E**) Violin plots of the number of stage 14 oocytes produced in *TOsk-Gal4 >w^RNAi* and *TOsk-Gal4 >otu^RNAi* flies. n=30. P-value < 0.001 (Student's t-test, unpaired, two-tailed) is indicated by ***.

The online version of this article includes the following source data and figure supplement(s) for figure 10:

**Source data 1.** Original uncropped gel blot images used in *Figure 10C*, *Figure 10—figure supplement 1*, indicating the relevant bands.

**Source data 2.** Original uncropped, unedited gel blot image files used in *Figure 10C*, *Figure 10—figure supplement 1*.

**Figure supplement 1.** Depletion of Sakura does not deplete ovarian tumor (Otu) and depletion of Otu does not deplete Sakura.

---

*1986*; *Storto and King, 1988*; *Steinhauer and Kalfayan, 1992*). To study the role of *otu* specifically in the germline, we performed RNAi knockdown of *otu* in the germline driven by nos-Gal4-VP16 (*Figure 10B and C*). Interestingly, we found that germline depletion of *otu* results in rudimentary ovaries, similar to the loss of *sakura* (*Figures 2C and 5A*, and *Figure 10B*). Furthermore, similar to *sakura* RNAi (*Figure 5*), *TOsk-Gal4*-driven *otu* RNAi knockdown did not affect ovary morphology (*Figure 10B*), but it resulted in a reduced number of eggs laid and stage 14 oocytes compared with control RNAi (w^RNAi) (*Figure 10D and E*).

Given that *sakura* loss of function causes tumorous and germless ovarioles, apoptosis, piRNA pathway defects, bam misexpression, and reduced pMad levels (*Figures 3, 4 and 7B–E*), we asked whether similar defects occur upon *otu* depletion. Germline *otu* RNAi knockdown in 2–5-days-old flies using UAS-Dcr2 and NGT-Gal4 leads to both germless (~70%) and tumorous (~30%) ovarioles, closely resembling the phenotype of *sakura* RNAi (*Figure 7—figure supplement 2*). Similar germless and tumorous phenotypes were also observed in *nos-Gal4-VP16>otu^RNAi* flies (*Figure 11A*). In these *otu^RNAi* ovarioles, Bam-GFP expression was no longer restricted to 8 cell cysts, instead persisting throughout the germarium and egg chambers (*Figure 11A*). Additionally, *otu^{14/14}* mutant GSCs showed reduced pMad levels (*Figure 11B*), and cleaved Caspase-3 staining revealed elevated apoptosis in *otu* RNAi ovaries (*Figure 11C*). Germline depletion of *otu* resulted in elevated expression of GFP and β-Gal produced by the Burdock sensor, indicating loss of piRNA-mediated transposon silencing (*Figure 11D*).

Together, these data demonstrate that *otu* loss-of-function closely phenocopies *sakura* loss-of-function, including defects in germline maintenance and differentiation, *bam* misexpression, reduced pMad signaling, and compromised piRNA pathway activity. These findings, along with physical interaction between Sakura and Otu (*Figures 8 and 9*, *Figure 8—figure supplement 1*, *Figure 9—figure supplement 1*), suggest that they function together in regulating germline maintenance and differentiation.

To further explore their genetic interaction, we compared the ovariole phenotypes caused by single and double knockdowns of *sakura* and *otu* in the germline driven by UAS-Dcr2 and NGT-Gal4. Individual knockdown of either gene resulted in ~70% germless and ~30% tumorous ovarioles in 2–5-days-old flies (*Figure 7—figure supplement 2*). Simultaneous knockdown of both genes increased the proportion of germless ovarioles to 86%, with a corresponding decrease in tumorous phenotypes to 14%, suggesting a synergistic enhancement of the germless phenotype. The number of GSC-like cells was not affected by the double knockdown of *sakura* and *otu* compared with their respective single knockdown (*Figure 7—figure supplement 2*).

## Sakura and Otu proteins do not depend on each other for their abundance

We investigated whether Sakura and Otu proteins depend on each other for their abundance. We performed Western blot analysis on ovaries with germline-specific knockdown of *sakura* or *otu*, driven by TOsk-Gal4. Otu protein levels were not reduced in *sakura^RNAi* ovaries compared to control *y^RNAi*, and conversely, Sakura levels were not decreased in *otu^RNAi* ovaries compared with control w^RNAi ovaries

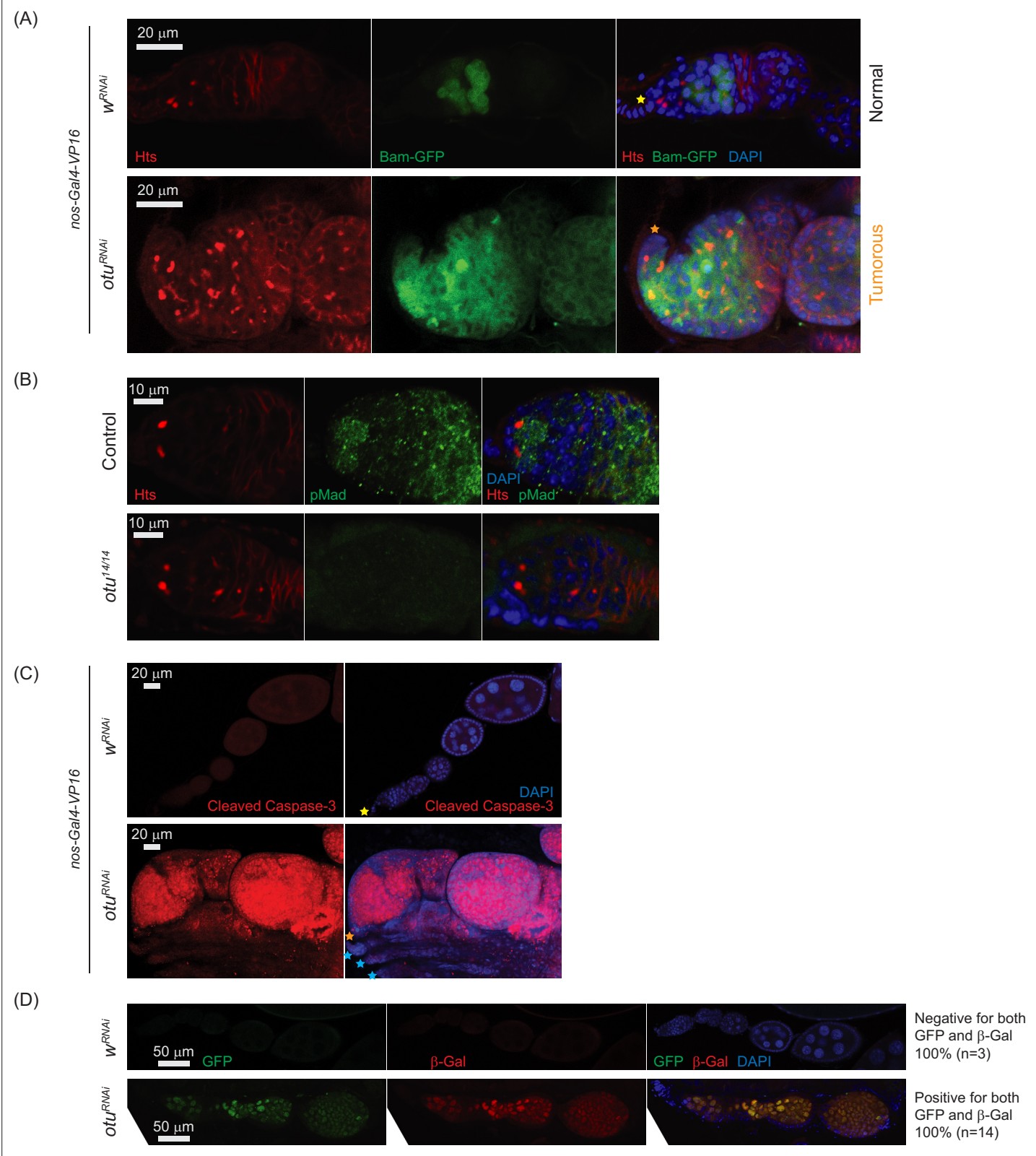

**Figure 11.** Loss of ovarian tumor (otu) results in low levels of phosphorylated Mad (pMad) and derepression of bag-of-marbles (bam) in the germaria. (**A**) Confocal images of the ovaries from control (*w*^RNAi^) and *otu*^RNAi^ flies harboring the *bam-GFP* reporter, where RNAi knockdown was driven in the female germline with nos-Gal4-VP16. Bam-GFP (green), Hts (red), and DAPI (blue). Scale bars: 20 μm. (**B**) Confocal images of germaria from control (*otu*^14/+^) and *otu*^14/14^ mutant flies stained with anti-pMad (green) and anti-Hts (red) antibodies. Scale bars: 10 μm. (**C**) Confocal images of the ovaries from

*Figure 11 continued on next page*

*Figure 11 continued*

*nos-Gal4-VP16>w*$^{RNAi}$ and *nos-Gal4-VP16>otu*$^{RNAi}$ flies stained with anti-cleaved Caspase 3 antibody. Cleaved Caspase-3 (red) and DAPI (blue). Scale bars: 20 µm. (**D**) Confocal images of the ovaries from *w*$^{RNAi}$ and *otu*$^{RNAi}$ flies carrying the Burdock sensor, with RNAi driven in the female germline using *UAS-Dcr2, NGT-Gal4*, and *nos-Gal4-VP16*. GFP (green), β-gal (red), and DAPI (blue). Scale bars: 50 µm. Three out of three tested control samples were negative for both GFP and β-gal, while 14 out of 14 tested *otu*$^{RNAi}$ samples were positive for GFP and β-gal.

(*Figure 10—figure supplement 1*). These results indicate that Sakura and Otu do not require each other for their protein expression or stability.

To further test this conclusion and to examine whether Otu enrichment in the posterior of egg chambers is affected by loss of *sakura*, we generated marked *sakura*$^{null}$ germline clones using the FLP-FRT system. Clones were marked by the absence of RFP, and experiments were performed in the flies expressing the *otu-EGFP* or *otu*(ΔTudor)-EGFP transgene in an otherwise wild-type (*otu*$^{+/+}$) background. Otu-EGFP fluorescence signal was comparable between *sakura*$^{null}$ clones (*FRT82B, sakura*$^{null}$) and control clones (*FRT82B*) (*Figure 12A*). Similarly, Otu(ΔTudor)-EGFP levels were unaffected in *sakura*$^{null}$ clones compared to the control clones (*Figure 12B*), confirming that Sakura is not required for Otu protein expression or stability.

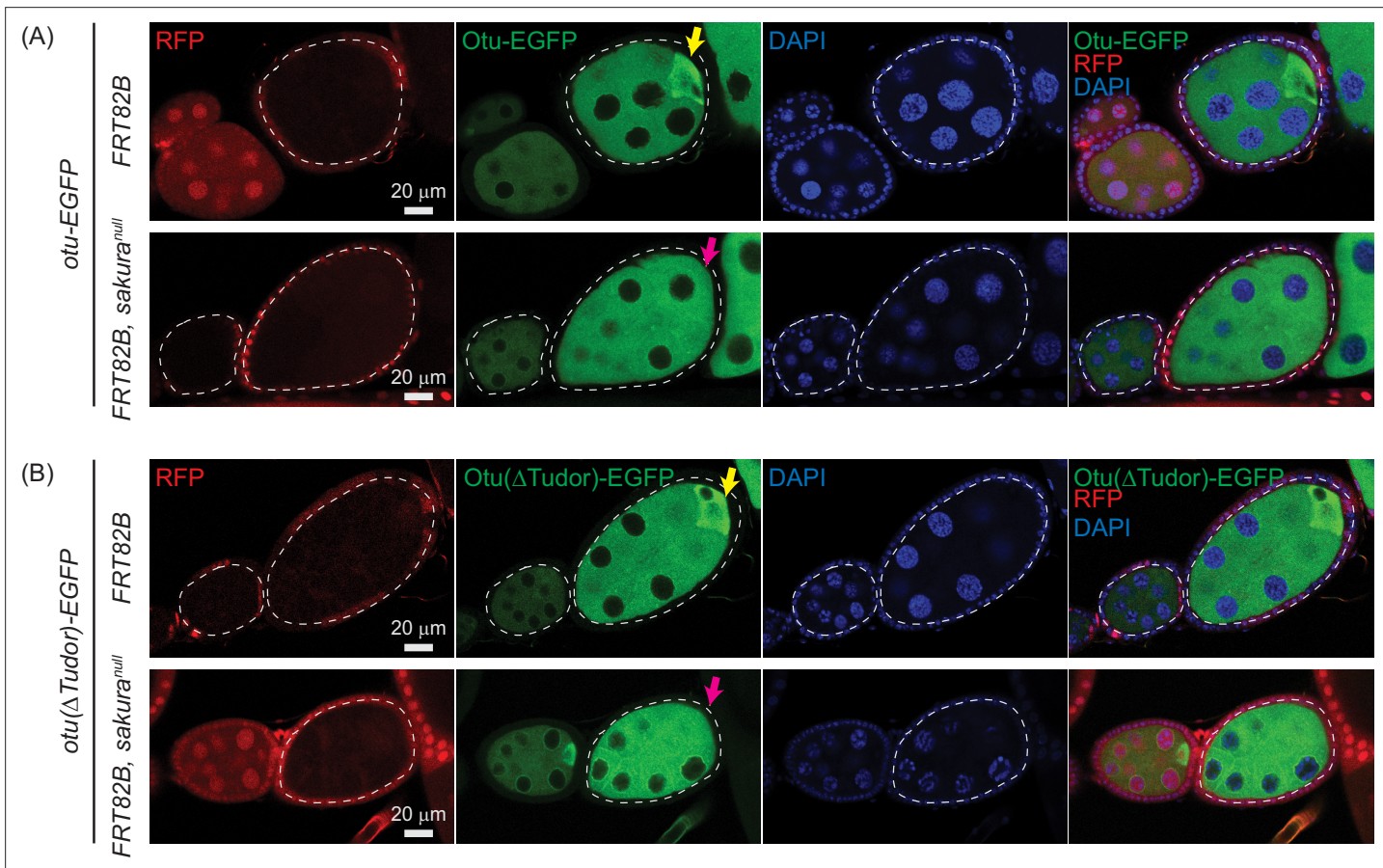

**Figure 12.** Otu enrichment to the posterior within egg chambers is lost in *sakura*$^{null}$. Confocal images of egg chambers with germline clones of *sakura*$^{null}$ expressing (**A**) Otu-EGFP or (**B**) Otu(ΔTudor)-EGFP. Fly genotypes used: *hs-flp/w; otu-EGFP/+; FRT82B, ubi-RFP/FRT82B. hs-flp/w; otu-EGFP/+; FRT82B, ubi-RFP/FRT82B, sakura*$^{null}$. *hs-flp/w; otu*(ΔTudor)-EGFP/+; FRT82B, ubi-RFP/FRT82B. hs-flp/w; otu(ΔTudor)-EGFP /+; FRT82B, ubi-RFP/FRT82B, sakura*$^{null}$. RFP (red), Otu-EGFP or Otu(ΔTudor)-EGFP (green), and DAPI (blue). Scale bars: 20 µm. Marked clones (RFP-negative) are outlined with white dotted lines. Yellow arrows indicate normal posterior enrichment of Otu-EGFP and Otu(ΔTudor)-EGFP signal; magenta arrows indicate the loss of this posterior enrichment in *sakura*$^{null}$ clones.

The online version of this article includes the following figure supplement(s) for figure 12:

**Figure supplement 1.** In vitro deubiquitination assay.

Both Otu-EGFP and Otu(ΔTudor)-EGFP exhibited clear enrichment at the posterior of egg chambers in the control clones (FRT82B) —consistent with localization to the developing oocyte (*Figure 12A and B*). This posterior enrichment was absent in *sakura^null* clones, demonstrating that Sakura is crucial for this process.

## Sakura does not affect Otu's deubiquitinase activity in vitro

A previous study has shown that Otu possesses deubiquitinase activity (*Ji et al., 2017*). We asked whether Sakura regulates the Otu's deubiquitinase activity. To test this, we performed an in vitro deubiquitination assay using Ub-Rhodamine 110 as a model substrate. Consistent with the previous study (*Ji et al., 2017*), we detected a deubiquitinase activity in Otu that was ectopically expressed and purified from S2 cells (*Figure 12—figure supplement 1*). The presence of Sakura, purified from *E. coli*, did not affect Otu's deubiquitinase activity under these assay conditions. Additionally, Sakura itself did not exhibit deubiquitinase activity.

## Discussion

In this study, we identified Sakura, encoded by a previously uncharacterized gene CG14545, as an essential factor for oogenesis and female fertility. We demonstrated that Sakura is specifically expressed in germline cells in the ovary, including GSCs, is localized to the cytoplasm, and is enriched in the developing oocytes. *sakura* homozygous null mutant flies are viable but completely female-sterile and male-fertile. Loss of *sakura*, either through null mutation or germline RNAi including in GSCs, results in rudimentary ovaries exhibiting germless and tumorous phenotypes.

The tumorous phenotype associated with the loss of *sakura* is characterized by an excess of GSC-like cells, which feature round spectrosomes, as well as cyst cells with branched fusomes (*Figure 3C*). The increased number of GSC-like cells in the loss of *sakura* suggests dysregulation of GSC self-renewal and differentiation. Meanwhile, the presence of excess cyst cells with branched fusomes indicates abnormal differentiation and division of cysts. Differentiation of cystoblasts typically involves four mitotic divisions with incomplete cytokinesis, leading to 16 cyst cells interconnected by branched fusomes. The degree of fusome branching serves as a marker for the stages of cyst cell division, with increased branching from 2-cell to 16-cell cysts (*de Cuevas and Spradling, 1998*). Notably, fusomes begin to degenerate and disappear after 16 cell cysts as the germline cyst enters the meiotic zone or region 2 of the germarium. The persistence of cyst cells with branched fusomes in *sakura^{null/null}* suggests that *sakura* is crucial for the proper division and differentiation of cysts, and we speculate that it is required for germline cysts to enter meiotic division.

Mosaic analysis of *sakura^null* indicates that *sakura* is intrinsically required for the establishment of GSCs in the ovary. When mutant clones were induced in the PGC stage, significantly fewer *sakura^null* marked GSCs were observed in adult ovaries, suggesting that many *sakura^null* PGCs fail to survive or differentiate into GSCs. Furthermore, induction of *sakura^null* clones in adult ovaries led to a more rapid decline in marked *sakura^null* GSCs compared to controls, indicating that *sakura* is also intrinsically required for GSC maintenance (*Figure 6A*). Over time, germaria containing *sakura^null* GSC clones became increasingly tumorous, with an expanding population of *sakura^null* GSC-like cells (*Figure 6B and C*). In contrast, the number of GSC-like cells across the entire and all ovarioles, regardless of whether GSCs in the niche remain, declined over time in *sakura^null* ovaries (*Figure 3—figure supplement 2*). These results suggest that *sakura^null* GSCs initially undergo aberrant proliferation, leading to tumor formation, but ultimately *sakura^null* GSCs and GSC-like cells are lost through a cell death mechanism, as evidenced by the elevated levels of cleaved Caspase-3 in *sakura^null* ovaries (*Figure 3F*).

Dpp/BMP signaling governs GSC self-renewal and cystoblast differentiation by repressing *bam* in GSCs and de-repressing it in daughter cystoblasts (*Figure 7A*). This process is mediated by the transcription factor Mad, which, when phosphorylated (pMad), translocates into the nucleus to repress bam transcription (*Kirilly and Xie, 2007*; *Kahney et al., 2019*; *Hinnant et al., 2020*). Our findings indicate that Bam expression is not limited to 8 cell cysts but continues throughout the germarium in ovaries lacking *sakura* function (*Figure 7B and C*). The misexpression of Bam in GSCs likely results from reduced levels of pMad in the GSCs (*Figure 7D and E*). Given the low pMad levels observed upon loss of *sakura*, we propose that *bam* misregulation in GSCs occurs primarily at the transcriptional level, although Bam is also known to be subject to post-transcriptional regulation (*Pek et al., 2009*).

The continued expression of Bam throughout the germarium, well beyond the GSC niche, suggests a failure to terminate *bam* expression at the 16 cell cyst stage — a developmental transition when *bam* is normally turned off. Thus, *sakura* mutants likely exhibit two distinct molecular defects contributing to *bam* overexpression: (1) impaired Dpp/BMP signaling within the GSC niche, and (2) a failure to downregulate *bam* at 16 cell stage. Given the limited space within the GSC niche, the latter defect may be the predominant contributor to the observed *bam* overexpression. The molecular mechanism that silences bam at the 16 cell cyst stage remains poorly understood, and how *sakura* loss perturbs this regulation is unclear. The sakura mutant may, therefore, serve as a valuable model to investigate the mechanism that normally suppresses *bam* expression at the 16 cell stage.

Transposons are mobile genetic elements that, if not silenced, can generate DNA damage and genomic instability (*Levin and Moran, 2011*). piRNAs derived from transposons and other repeats can target and silence transposon RNAs to preserve genome integrity in germ cells (*Siomi et al., 2011*). Loss of piRNAs leads to transposon derepression, resulting in increased DNA damage, which subsequently triggers cell death (*Kang et al., 2018*; *Moon et al., 2018*). Genetic damage in germ cells can cause developmental defects and diseases that may be inherited by the next generation. Thus, the elimination of defective germ cells is crucial for maintaining germline integrity of a species (*Chu et al., 2014*; *Ota and Kobayashi, 2020*). We found that loss of *sakura* results in reduced piRNA levels and loss of piRNA-mediated transposon silencing in the germline (*Figure 4*). The observed apoptosis, indicated by elevated cleaved Caspase-3 levels in *sakura* mutant ovaries (*Figure 3F*), suggests that desilencing of transposons due to reduced piRNA levels likely results in increased DNA damage, triggering cell death. We speculate that the germless phenotype in *sakura* mutants may partly arise from an apoptotic germline elimination program activated to maintain germline integrity.

In addition to its expression in GSCs and cysts in the germarium, Sakura is also expressed in germline cells in later-stage egg chambers (*Figure 1E–F*) and is required at this stage for proper oogenesis. When Sakura is depleted from region 2b of the germarium onward —leaving GSCs and early cyst cells intact—females have significantly fewer stage 14 oocytes and lay significantly fewer eggs (*Figure 5*). The failure to produce stage 14 mature oocytes likely stems from cytoskeletal disorganization that disrupts oocyte development, as evidenced by mislocalization of Orb (*Figure 5*, *Figure 5—figure supplement 1*, *Figure 7—figure supplement 1*). Notably, piRNA pathway mutants also exhibit similar Orb mislocalization (*Ohtani et al., 2013*). Therefore, the reduced piRNA levels (*Figure 4*) may contribute to the oocyte development defects, including Orb mislocalization, in *sakura* mutants (*Figure 5E*).

Sakura does not possess any known protein domains. To infer its function, we identified Otu, a protein known to be crucial for oogenesis, as a protein partner of Sakura. Mutations in *otu* gene lead to a range of ovarian phenotypes, including germ cell loss, tumorous egg chambers filled with undifferentiated germ cells, defects in germline sexual identity determination, abnormalities in nurse cell chromosome structure, and defects in oocyte determination (*King and Riley, 1982*; *Storto and King, 1988*; *Pauli et al., 1993*; *Glenn and Searles, 2001*). We showed that germline depletion of Otu via RNAi phenocopies loss of *sakura*. Similar to Sakura, loss of *otu* inhibits Dpp/BMP signaling, resulting in low pMad levels in GSCs and Bam de-repression (*Figure 11A–B*). Additionally, loss of *otu* also results in the loss of piRNA-mediated silencing, paralleling the effects of *sakura* loss. Furthermore, germline knockdown of *otu* yielded a similar ratio of germless ovaries as seen in *sakura*-RNAi, and simultaneous knockdown of both *otu* and *sakura* exacerbated the germless ovary phenotype (*Figure 7—figure supplement 2*). These observations raise the possibility that Sakura and Otu function together to regulate germ cell maintenance in the ovaries and are involved in Dpp/BMP signaling to balance stem cell renewal and differentiation.

Otu possesses deubiquitinase activity, catalyzed by its N-terminal Otu domain (*Ji et al., 2017*; *Ji et al., 2019*). In germ cells, Otu interacts with Bam, forming a deubiquitinase complex that deubiquitinates and thereby stabilizes CycA, promoting GSC differentiation. The predicted structure of Sakura and Otu complex suggests that the Mid domain of Sakura directly contacts the Otu domain of Otu (*Figure 9A*). We found that Sakura lacks deubiquitinase activity and it does not directly affect Otu's deubiquitinase rate in our in vitro assays using Ub-Rhodamine 110 as a substrate (*Figure 12—figure supplement 1*). This does not preclude the possibility that Sakura may still influence Otu's deubiquitinase activity, potentially guiding Otu to its substrate and determining substrate specificity. Our in vitro assays may not have detected such specificity changes.

Otu is an RNA-binding protein whose deubiquitinase activity is enhanced by RNA binding (*Ji et al., 2019*). Bam, along with other proteins such as Bgcn, Mei-P26, and Sxl, binds nanos mRNA —a key stem cell maintenance factor (*Wang and Lin, 2004*)— and represses its translation once germ cells have exited the GSC niche (*Wang and Lin, 2004*; *Li et al., 2009*; *Chau et al., 2012*; *Li et al., 2013*). Bam and Otu form a protein complex (*Ji et al., 2017*). Future studies should explore whether Sakura modulates Otu's RNA-binding properties and its interaction with other proteins. Although Sakura is exclusively expressed in ovaries, particularly in germline cells including GSCs (*Figure 1*), Otu is broadly expressed in various tissues, including testes and gut (*Steinhauer and Kalfayan, 1992*). This restricted expression pattern suggests that Sakura may serve as a female germline-specific cofactor that enhances or modifies Otu's molecular functions. Identifying Otu's RNA targets and deubiquitinase substrates beyond CycA, as well as determining how Sakura binding influences these activities, will be key to elucidating their roles in oogenesis. One intriguing possibility is that Sakura regulates Otu's activity toward piRNA pathway components, thereby contributing to proper piRNA biogenesis and function. Additionally, the Sakura-Otu complex may directly regulate essential post-transcriptional processes such as *sxl* alternative splicing and translational control of other oogenic RNAs.

In summary, this study identifies and characterizes the previously unknown gene *sakura*, which is specifically expressed in female germ cells and is essential for oogenesis and female fertility. Together with its protein partner Otu, Sakura likely regulates germline cell fate, maintenance, and differentiation.

## Materials and methods

### Fly strains

We generated the *sakura^null^* strain by introducing indels within the Sakura coding region using the CRISPR/Cas9 genome editing system, as we previously reported (*Zhu et al., 2018a*; *Zhu et al., 2018b*; *Zhu et al., 2019b*; *Zhu and Fukunaga, 2021*). The transgenic *sakura-EGFP, otu-EGFP,* and otu(ΔTudor)-EGFP strains were created following previously published methods (*Fukunaga et al., 2012*; *Kandasamy and Fukunaga, 2016*; *Zhu and Fukunaga, 2021*). A fragment containing Sakura cDNA flanked by ~1 kbp upstream and ~1 kbp downstream genomic sequences was cloned. Fragments containing Otu and Otu(ΔTudor) cDNAs, flanked by ~2.5 kbp upstream and ~1 kbp downstream genomic sequences, were also cloned. The EGFP gene was fused in-frame to the C-terminus of the Sakura and Otu coding sequences within these fragments, which were then inserted into a pattB plasmid vector. The *sakura-EGFP* plasmid was integrated into the 51C1 site within the fly genome using the BDSC:24482 fly strain and the PhiC31 system, while the *otu-EGFP* and otu(ΔTudor)-EGFP plasmids were integrated into the 25C6 site of the genome using the attP40 fly strain and the PhiC31 system.

The *sakura RNAi* #1 (VDRC: v39727) and #2 (VDRC: v103660), y-RNAi (v106068), TOsk-Gal4 (v 314033), Burdock sensor [*UAS-Dcr2; NGT-Gal4; nosGal4-VP16, nos >NLS_GF'_lacZ_vas-3'UTR_burdock-target*] (v 313217) strains were from the Vienna *Drosophila* Resource Center. w-RNAi (BDSC: 35573), *otu*-RNAi (BDSC: 34065), *bam*-RNAi (BDSC: 33631), cycA-RNAi (BDSC: 29313), *UAS-Dcr-2; NGT-Gal4* (BDSC: 25751), *FRT82B/TM6C, Sb* (BDSC: 86313), *otu^14^* (BDSC: 6025), UASp-Mad-GFP (BDSC: 604592), and UASp-tkv.Q253D (BDSC: 604934) were obtained from the Bloomington Stock Center. The *bam-GFP* reporter (DGRC: 118177) and vasa-EGFP knocked-in fly (DGRC: 118616) were from the Kyoto *Drosophila* Stock Center. *sakura RNAi #2* was much healthier than *sakura RNAi #1* without being driven with any Gal4 for some reason, and thus we chose to use the RNAi #2 line in most of this study. sakura-EGFP does not cause any phenotypes in the background of *sakura^+/+^* and *sakura ^null/+^*.

### Fertility assay

The female fertility assay was performed as previously described (*Zhu et al., 2018b*; *Liao et al., 2019*; *Zhu et al., 2019a*; *Zhu and Fukunaga, 2021*). Briefly, five test virgin females of each test genotype were mated with three wild-type (OregonR) males in a cage with a 6 cm grape juice agar plate supplemented with wet yeast paste. The agar plates were replaced daily, and the number of eggs laid on the third plate (on day 3) was recorded. After incubation at 25 °C for an additional day, the number of hatched eggs was counted. At least three cages per genotype were tested.

For the male fertility assay, a single test male was mated with five wild-type (OregonR) virgin females in each vial, following previously published methods (*Zhu et al., 2018b*; *Zhu et al., 2019a*; *Zhu and Fukunaga, 2021*). After 3 days, the females were transferred to a new vial (vial 1). Every two days, they were transferred to a new vial until a total of four vials were obtained. After 2 days in the fourth vial, the females were removed, and the total number of progenies emerging from these four vials was counted. At least five males per genotype were tested.

## Sakura and Otu antibodies

We expressed a recombinant full-length Sakura protein as an N-terminal 6xHis-tagged protein in *E. coli* using a modified pET vector and purified it using Ni-sepharose (GE Healthcare) and HiTrapQ HP (GE Healthcare) columns (*Fukunaga and Doudna, 2009*). Recombinant Otu fragments (145-405aa and 406-853aa) were similarly expressed as N-terminally 6xHis-MBP-fusion proteins in *E. coli* and purified using Ni-sepharose. These purified proteins were used as antigens to generate polyclonal anti-Sakura and anti-Otu sera in rabbits (Pocono Rabbit Farm & Laboratory, Inc). The rabbit polyclonal anti-Sakura antibodies were affinity purified using His-MBP-Sakura recombinant protein and Affigel-15 (Bio-Rad), following the manufacturer's instructions. Rabbit anti-Otu sera were first pre-cleared with purified His-MBP protein bound to Affigel-15, and the unbound fraction was affinity purified using purified His-MBP-Otu fragments (145-405aa and 406-853aa) and Affigel-15.

## Immunostaining

Stereomicroscope images of dissected ovaries were taken using Leica M125 stereomicroscope. Ovaries from 2–5-day-old, yeast-fed females were hand-dissected in 1 X PBS (137 mM NaCl, 2.7 mM KCl, 10 mM $Na_2HPO_4$, 1.8 mM $KH_2PO_4$, pH 7.4) at room temperature. The dissected ovaries were fixed in a fixative buffer (4% formaldehyde, 15 mM PIPES (pH 7.0), 80 mM KCl, 20 mM $NaCl_2$, 2 mM EDTA, and 0.5 mM EGTA), incubated for 30 min at room temperature with gentle rocking. After fixation, the ovaries were rinsed three times with PBX (0.1% Triton X-100 in 1 X PBS) and then incubated in a blocking buffer (2% donkey serum in 3% BSA [w/v], 0.02% $NAN_3$ [w/v] in PBX) for 1 hr at room temperature. Then, the ovaries were incubated with primary antibodies diluted in the blocking buffer overnight at 4 °C. The following day, the ovaries were rinsed three times with PBX and incubated with Alexa Fluor-conjugated secondary antibodies for 2 hr. The ovaries were rinsed three times with PBX and then mounted in VECTASHIELD PLUS antifade mounting medium with DAPI (H-2000, Vector lab). Confocal images were acquired on a Zeiss LSM700 confocal microscope at the Johns Hopkins University School of Medicine Microscope Facility.

The primary antibodies used for immunostaining were mouse anti-HTS (1B1) (DSHB, AB_528070, dilution: 1/100), mouse anti-Bam (DSHB, AB_10570327, 1/20), mouse anti-CycA (DSHB, AB_528188, 1/100), rat anti-Vasa (DSHB, AB_760351, 1/100), rat anti-DE Cadherin (DCAD2) (DSHB, AB_528120, 1/20), rabbit anti-pMad (Cell Signaling, Phospho-SMAD1/5 (Ser463/465) mAb #9516, 1/200), and rabbit anti-cleaved caspase-3 (Cell Signaling, Cleaved Caspase-3 (Asp175) #9661, 1/200). Secondary antibodies used were Alexa Fluor 488 Donkey anti-Mouse Igg (Thermo Fisher, A21202, 1/100), Alexa Fluor 594 Donkey anti-Rat Igg (Thermo Fisher, A21209, 1/100), Alexa Fluor 594 Donkey anti-Mouse Igg (Thermo Fisher, A21203, 1/100), and Alexa Fluor 594 Donkey anti-Rabbit Igg (Thermo Fisher, A21207, 1/100). Rhodamine phalloidin (Thermo Fisher, R415, 1/100) was used to stain F-Actin.

While our Sakura antibody detects Sakura in immunostaining, it seems to detect some other proteins as well. Since we have Sakura-EGFP fly strain, which rescues *sakura^null* phenotypes, we relied on Sakura-EGFP rather than anti-Sakura antibodies immunostaining to examine Sakura expression and localization.

## Germline clonal analysis

We generated *sakura^null* mutant clones using FLP/FRT-mediated recombination (*Rubin and Huynh, 2015*). To induce *sakura^null* GSC clones, 3-day-old female flies of the genotype hs-flp/w; +; FRT82B, ubi-GFP/FRT82B, sakura^null were heat-shocked at 37 °C for 1 hr, twice daily, with an 8 hr interval between heat shocks. Female flies of the genotype hs-flp/w; +; FRT82B, ubi-GFP/FRT82B were used as controls. Ovaries were dissected and stained 4, 7, and 14 days after clone induction. To induce PGC clones, early third-instar larvae were subjected to the same heat shock regime, then allowed to develop to adulthood. Ovaries were dissected and analyzed at 3–4 days after eclosion.

To induce *sakura^null* mutant clones in the presence of *otu-EGFP* or *otu(ΔTudor)-EGFP* transgenes, 3-day-old female flies with the genotype *hs-flp/w*; *otu-EGFP/+*; *FRT82B*, ubi-RFP/*FRT82B*, *sakura^null* and *hs-flp/w*; *otu(ΔTudor)-EGFP /+*; *FRT82B*, ubi-RFP/*FRT82B*, *sakura^null* were heat-shocked under the same conditions. Flies with the genotypes of *hs-flp/w*; *otu-EGFP/+*; *FRT82B*, ubi-RFP/*FRT82B* and *hs-flp/w*; *otu(ΔTudor)-EGFP/+*; *FRT82B*, ubi-RFP/*FRT82B* were used as controls. Flies were dissected 3–4 days after clone induction.

## Western blot

Lysates of hand-dissected ovaries and tissues were prepared by homogenizing in RIPA buffer (50 mM Tris-HCl [pH 7.4], 150 mM NaCl, 1% [v/v] IGEPAL CA-630, 0.1% [w/v] sodium dodecyl sulfate (SDS), 0.5% [w/v] sodium deoxycholate, 1 mM ethylenediaminetetraacetic acid (EDTA), 5 mM dithiothreitol, and 0.5 mM phenylmethylsulfonyl fluoride (PMSF)) (*Kandasamy et al., 2017*; *Zhu et al., 2018b*). The homogenates were centrifuged at 21,000 g at 4 °C for 10 min, and the protein concentration of the supernatant was determined using the BCA protein assay kit (Pierce) as needed. Fifteen μg of total protein was loaded per lane for Western blot. The sources and dilutions of the primary antibodies were as below. Rabbit anti-Sakura (1/10,000, generated in this study), rabbit anti-Otu (1/10,000, generated in this study), rabbit anti-alpha-Tubulin [EP1332Y] (1/10,000, Abcam, ab52866), mouse anti-alpha-Tubulin [12G10] (1/10,000, DSHB, AB_1157911), mouse anti-FLAG (1/10,000, Sigma, F1804), mouse anti-HA (1/10,000, Sigma, H3663), and mouse anti-GFP [GF28R] (1/3000, Invitrogen, 14-6674-82). IRDye 800CW goat anti-mouse IgG, IRDye 800CW goat anti-rabbit IgG, IRDye 680RD goat anti-mouse, and IgG IRDye 680RD goat anti-rabbit were used as secondary antibodies. The membranes were scanned using the Li-Cor Odyssey CLx Imaging System.

## Mass spectrometry

Immunoprecipitation of Sakura-EGFP protein was performed using the GFP-Trap Magnetic Agarose Kit (Proteintech, gtmak-20) on dissected ovaries from flies harboring *sakura-EGFP* transgene, with w1118 flies as controls. Ovaries were homogenized in 200 μL ice-cold lysis buffer (10 mM Tris-HCl [pH 7.5], 150 mM NaCl, 0.5 mM EDTA, 0.05% [v/v] IGEPAL CA-630) containing 1x protease inhibitor cocktail (100x protease inhibitor cocktail contains 120 mg/ml 1 mM 4-(2-aminoethyl) benzene sulfonyl fluoride hydrochloride (AEBSF), 1 mg/ml aprotinin, 7 mg/ml bestatin, 1.8 mg/ml E-64, and 2.4 mg/ml leupeptin). After homogenization, the tubes were placed on ice for 30 min, and the homogenates were extensively pipetted every 10 min. The lysates were then centrifuged at 17,000 x g for 10 min at 4 °C. The supernatants were transferred to pre-chilled tubes, and 300 μL dilution buffer (10 mM Tris/Cl pH 7.5, 150 mM NaCl, 0.5 mM EDTA) supplemented with 1 x protease inhibitor cocktail were added. The diluted lysates were then added to the GFP-trap magnetic beads in 1.5 mL tubes and rotated for 1 hr at 4 °C. After separating the beads with a magnetic tube rack, the beads were washed three times with 500 μL wash buffer (10 mM Tris/Cl pH 7.5, 150 mM NaCl, 0.05% [v/v] IGEPAL CA-630). Proteins were eluted with 40 μL acidic elution buffer (200 mM glycine pH 2.5) followed by immediate neutralization with 5 μL neutralization buffer (1 M Tris pH 10.4).

As a quality control before mass spectrometry, ~5 μL of the samples were mixed with an equal volume of 2x SDS PAGE loading buffer (80 mM Tris-HCl [pH 6.8], 2% [w/v] SDS, 10% [v/v] glycerol, 0.0006% [w/v] bromophenol blue, 2% [v/v] 2-mercaptoethanol), heated at 95 °C for 3 min, and run on 4–20% Mini-PROTEAN TGX Precast Protein Gels (Bio-Rad, #4561094). Silver staining was then performed by using the Pierce Silver Stain Kit (Thermo Fisher, 24612) to assess the quality of the immunoprecipitated protein samples. Mass spectrometry was conducted at the Mass Spectrometry Core at the Department of Biological Chemistry, Johns Hopkins School of Medicine, as previously described (*Zhu and Fukunaga, 2021*).

## Co-immunoprecipitation

For co-immunoprecipitation of endogenous Sakura protein, wild-type ovaries (w1118) were homogenized in ice-cold lysis buffer (10 mM Tris-HCl [pH 7.5], 150 mM NaCl, 0.5 mM EDTA, 0.05% [v/v] IGEPAL CA-630) containing 1×protease inhibitor cocktail, centrifuged at 21,000 g at 4 °C for 10 min, and the clear supernatant protein lysates were used for immunoprecipitation. Four μg of rabbit anti-Sakura and normal rabbit IgG (Cell Signaling, #2729) were incubated with 50 μL of Dynabeads Protein G (Thermo Fisher, 10004D) for 20 min at room temperature. The beads were washed once with PBST

(1 x PBS with 0.1% Tween-20). The ovary lysate supernatant was then incubated with the washed beads at room temperature for 30 min, followed by three washes with PBST. The proteins were eluted with 2 x SDS-PAGE loading buffer (80 mM Tris-HCl [pH 6.8], 2% [w/v] SDS, 10% [v/v] glycerol, 0.0006% [w/v] bromophenol blue, 2% [v/v] 2-mercaptoethanol) and heated at 70 °C for 10 min. After bead separation using a magnetic tube rack, the eluted proteins in 2 x SDS-PAGE loading buffer were heated again at 95 °C for 3 min.

Transient protein expression in S2 cells was performed using the pAc5.1/V5-HisB plasmid vector (Invitrogen). A total of 1 µg plasmids were transfected using the Effectene transfection reagent (Qiagen, 301425). Three days after transfection, cells were harvested and lysed with ice-cold lysis buffer supplemented with 1x protease inhibitor cocktail. The cell lysates were then centrifuged at 17,000 x g for 10 min at 4 °C, and the clear supernatants were collected for immunoprecipitation. For anti-HA immunoprecipitation, supernatants were incubated with 25 µL (0.25 mg) of Pierce Anti-HA Magnetic Beads (Thermo Fisher, 88837) at room temperature for 30 min. The beads were then washed three times with TBST (1 x TBS [0.05 M Tris/HCl and 0.15 M NaCl, pH 7.6] with 0.05% Tween-20). For anti-FLAG immunoprecipitation, 2 µg of mouse anti-FLAG (1/10,000, Sigma, F1804) was incubated with 50 µL of Dynabeads Protein G (Thermo Fisher, 10004D) for 10 min at room temperature. The beads were washed once with PBST. The S2 cell lysate supernatant was incubated with the washed beads at room temperature for 15 min. The beads were washed three times with PBST. In both anti-HA and anti-FLAG immunoprecipitations, proteins were eluted with 2x SDS PAGE loading buffer. For anti-HA, the beads in 2 x SDS-PAGE loading buffer were heated at 95 °C for 7 min. For anti-FLAG, the beads were heated at 70 °C for 10 min, then were separated using a magnetic tube rack. The eluted proteins in 2 x SDS-PAGE loading buffer were heated again at 95 °C for 3 min.

While the anti-Otu antibody worked in immunoprecipitation from ovaries, in subsequent anti-Sakura Western blot after anti-Otu IP, Otu antibody light chain bands often overlapped with the Sakura band. Using epitope tags in S2 cells avoided this issue.

## Small RNA and mRNA sequencing

Poly-A +mRNA purification was performed as previously described (*Fukunaga et al., 2012*). Small RNA libraries and poly-A +mRNA libraries were prepared, sequenced on Hiseq2500 (Illumina), and analyzed, as previously reported (*Fukunaga et al., 2014*; *Kandasamy et al., 2017*; *Liao et al., 2018*; *Zhu et al., 2018a*; *Zhu et al., 2018b*; *Liao et al., 2019*; *Zhu et al., 2019a*; *Zhu et al., 2019b*). SRA accession number for these datasets is PRJNA1156618.

## RT-PCR

Total RNAs from testes and ovaries were prepared using miRVana (Thermo Fisher Scientific). RNAs were treated with Turbo DNase (Thermo Fisher Scientific) to remove potential genomic DNA contamination. A total of 1 µg of RNA was reverse-transcribed into cDNA using SuperScript VILO MasterMix (Thermo Fisher Scientific). To assess *sxl* alternative splicing, PCR was performed using GoTaq Green Master Mix (Promega) with the primers *sxl*-F (CTCACCTTCGATCGAGGGTGTA) and *sxl*-R (GATG GCAGAGAATGGGAC).

## In vitro deubiquitination assay

C-terminally 6xHis-tagged recombinant Sakura protein was expressed in *E. coli* using a modified pET vector and was purified using Ni-sepharose. N-terminally 3xHA-HRV3Csite-3xFLAG-tagged Otu protein and a negative control, N-terminally 3xHA-HRV3Csite-3xFLAG-tagged firefly luciferase protein, were expressed in S2 cells using pAc5.1/V5-HisB vector. The proteins were immunoprecipitated using Pierce Anti-HA Magnetic Beads (Thermo Fisher, 88837) as described previously, with a more stringent washing step to effectively remove interacting proteins. The beads were washed six times with a high salt wash buffer (TBST with 800 mM NaCl). The proteins were eluted by cleaving off 3xHA tag with 25 nM GST-HRV3C protease in a cleavage buffer (25 mM Tris-HCl pH 7.4, 150 mM NaCl, 5% glycerol, 2 mM EDTA), incubated at 4 °C with rotation for 6 hr. An equal volume of 100% glycerol was added to the eluted protein.

Purified proteins, including Otu, luciferase, and Sakura, were mixed with Ub-Rhodamine 110 (Ubiquitin-Proteasome Biotechnologies, M3020) in a total volume of 30 µL in reaction buffer (20 mM Tris-HCl, pH 7.5, 200 mM NaCl, 5 mM MgCl$_2$, 2 mM DTT). The mixture was added to a black 384-well

low-volume plate. Fluorescence measurements were taken using the SpectraMax i3x Multi-Mode Microplate Reader at 37 °C, with excitation and emission wavelengths set at 485/20 and 530/20 nm, respectively. The fluorescence intensity for each condition was averaged from triplicates and plotted as a function of time.

## Materials availability statement

Materials newly created in this study, including plasmids and fly strains, are available from the authors upon request.

## Acknowledgements

We thank the Bloomington *Drosophila* Stock Center, the Vienna *Drosophila* Resource Center, and the Kyoto *Drosophila* Stock Center for fly strain stocks. We thank the Johns Hopkins University School of Medicine Microscope Facility for use of the Zeiss LSM700. We thank Ms. Lauren DeVine and Dr. Bob Cole at the Johns Hopkins University School of Medicine Mass Spectrometry and Proteomics Core Facility for the mass-spec analysis. This work was supported by the grants from the National Institutes of Health [R35GM145352 and R03AI178064] and Johns Hopkins University Catalyst Award to RF.

## Additional information

### Funding

| Funder | Grant reference number | Author |
| --- | --- | --- |
| National Institute of General Medical Sciences | R35GM145352 | Ryuya Fukunaga |
| National Institute of Allergy and Infectious Diseases | R03AI178064 | Ryuya Fukunaga |
| Johns Hopkins University Catalyst Award | | Ryuya Fukunaga |

The funders had no role in study design, data collection and interpretation, or the decision to submit the work for publication.

### Author contributions

Azali Azlan, Conceptualization, Investigation, Methodology, Writing – original draft, Writing – review and editing; Li Zhu, Investigation, Methodology; Ryuya Fukunaga, Conceptualization, Supervision, Funding acquisition, Investigation, Methodology, Writing – review and editing

### Author ORCIDs

Azali Azlan ⓘ http://orcid.org/0000-0001-6474-4875
Ryuya Fukunaga ⓘ https://orcid.org/0000-0002-5814-8206

Reviewer #1 (Public review): https://doi.org/10.7554/eLife.103828.4.sa1
Reviewer #2 (Public review): https://doi.org/10.7554/eLife.103828.4.sa2
Reviewer #3 (Public review): https://doi.org/10.7554/eLife.103828.4.sa3
Author response https://doi.org/10.7554/eLife.103828.4.sa4

## Additional files

### Supplementary files

MDAR checklist

### Data availability

Sequencing data have been deposited in SRA under accession number PRJNA1156618.

The following dataset was generated:

| Author(s) | Year | Dataset title | Dataset URL | Database and Identifier |
|---|---|---|---|---|
| Ryuya F | 2024 | Sequencing RNAs from fruit fly ovaries | https://www.ncbi.nlm.nih.gov/bioproject/PRJNA1156618 | NCBI BioProject, PRJNA1156618 |

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
