## [Editor Report · eLife Assessment]

This **valuable** study reports the first characterization of the CG14545 gene in *Drosophila melanogaster*, which the authors name "Sakura." Acting during germline stem cell fate and differentiation, Sakura is required for both oogenesis and female fertility, although some mechanistic details require further investigation. This **solid** study presents a wide-ranging and well-controlled characterization of Sakura, and accordingly the findings and associated reagents described will be of use to scientists interested in oogenesis and early development.

---

## [Referee Report · Reviewer #1 (Public review)]

In this manuscript, Azlan et al. identified a novel maternal factor called Sakura that is required for proper oogenesis in *Drosophila*. They showed that Sakura is specifically expressed in the female germline cells. Consistent with its expression pattern, Sakura functioned autonomously in germline cells to ensure proper oogenesis. In sakura KO flies, germline cells were lost during early oogenesis and often became tumorous before degenerating by apoptosis. In these tumorous germ cells, piRNA production was defective and many transposons were derepressed. Interestingly, Smad signaling, a critical signaling pathway for the GSC maintenance, was abolished in sakura KO germline stem cells, resulting in ectopic expression of Bam in whole germline cells in the tumorous germline. A recent study reported that Bam acts together with the deubiquitinase Otu to stabilize Cyc A. In the absence of sakura, Cyc A was upregulated in tumorous germline cells in the germarium. Furthermore, the authors showed that Sakura co-immunoprecipitated Otu in ovarian extracts. A series of in vitro assays suggested that the Otu (1-339 aa) and Sakura (1-49 aa) are sufficient for their direct interaction. Finally, the authors demonstrated that the loss of otu phenocopies the loss of sakura, supporting their idea that Sakura plays a role in germ cell maintenance and differentiation through interaction with Otu during oogenesis.

Latest comments:

The reviewer acknowledges the importance of sharing the observed defects in Sakura mutant ovaries and the possible physiological significance of the Sakura-Out interaction with the research community, as this information could lay the groundwork for future functional analysis research.

---

## [Referee Report · Reviewer #2 (Public review)]

In this study, the authors identified CG14545 (named it sakura), as a key gene essential for *Drosophila* oogenesis. Genetic analyses revealed that Sakura is vital for both oogenesis progression and ultimate female fertility, playing a central role in the renewal and differentiation of germ stem cells (GSC).

The absence of Sakura disrupts the Dpp/BMP signaling pathway, resulting in abnormal bam gene expression, which impairs GSC differentiation and leads to GSC loss. Additionally, Sakura is critical for maintaining normal levels of piRNAs. Also, the authors convincingly demonstrate that Sakura physically interacts with Otu, identifying the specific domains necessary for this interaction, suggesting a cooperative role in germline regulation. Importantly, the loss of otu produces similar defects to those observed in sakura mutants, highlighting their functional collaboration.

The authors provide compelling evidence that Sakura is a critical regulator of germ cell fate, maintenance, and differentiation in *Drosophila*. This regulatory role is mediated through modulation of pMad and Bam expression. However, the phenotypes observed in the germarium appear to stem from reduced pMad levels, which subsequently trigger premature and ectopic expression of Bam. This aberrant Bam expression could lead to increased CycA levels and altered transcriptional regulation, impacting piRNA expression. In this revised manuscript, the authors further investigated whether Sakura affects the function of Orb, a binding partner they identified, in deubiquitinase activity when Orb interacts with Bam.

This elaborate study will be embraced by both germline-focused scientists and the developmental biology community.

Latest comments:

The authors answered all my persistent concerns and made changes according to the recommendations I incorporated for the revised version of the manuscript.

---

## [Referee Report · Reviewer #3 (Public review)]

In this very thorough study, the authors characterize the function of a novel *Drosophila* gene, which they name Sakura. They start with the observation that sakura expression is predicted to be highly enriched in the ovary and they generate an anti-sakura antibody, a line with a GFP-tagged sakura transgene, and a sakura null allele to investigate sakura localization and function directly. They confirm the prediction that it is primarily expressed in the ovary and, specifically, that it is expressed in germ cells, and find that about 2/3 of the mutants lack germ cells completely and the remaining have tumorous ovaries. Further investigation reveals that Sakura is required for piRNA-mediated repression of transposons in germ cells. They also find evidence that sakura is important for germ cell specification during development and germline stem cell maintenance during adulthood. However, despite the role of sakura in maintaining germline stem cells, they find that sakura mutant germ cells also fail to differentiate properly such that mutant germline stem cell clones have an increased number of "GSC-like" cells. They attribute this phenotype to a failure in the repression of Bam by dpp signaling. Lastly, they demonstrate that sakura physically interacts with otu and that sakura and otu mutants have similar germ cell phenotypes. Overall, this study helps to advance the field by providing a characterization of a novel gene that is required for oogenesis. The data are generally high-quality and the new lines and reagents they generated will be useful for the field.

Latest comments:

As with my previous assessment, I remain supportive of publication of this manuscript. Though I agree with the other reviewers that additional experimentation would increase the value of this study even further, I feel it will also be a useful contribution to the field as is.

---

## [Author Response]

The following is the authors’ response to the previous reviews

**Public Reviews:**

**Reviewer #1 (Public review):**
Summary:In this manuscript, Azlan et al. identified a novel maternal factor called Sakura that is required for proper oogenesis in *Drosophila*. They showed that Sakura is specifically expressed in the female germline cells. Consistent with its expression pattern, Sakura functioned autonomously in germline cells to ensure proper oogenesis. In sakura KO flies, germline cells were lost during early oogenesis and often became tumorous before degenerating by apoptosis. In these tumorous germ cells, piRNA production was defective and many transposons were derepressed. Interestingly, Smad signaling, a critical signaling pathway for the GSC maintenance, was abolished in sakura KO germline stem cells, resulting in ectopic expression of Bam in whole germline cells in the tumorous germline. A recent study reported that Bam acts together with the deubiquitinase Otu to stabilize Cyc A. In the absence of sakura, Cyc A was upregulated in tumorous germline cells in the germarium. Furthermore, the authors showed that Sakura co-immunoprecipitated Otu in ovarian extracts. A series of in vitro assays suggested that the Otu (1-339 aa) and Sakura (1-49 aa) are sufficient for their direct interaction. Finally, the authors demonstrated that the loss of otu phenocopies the loss of sakura, supporting their idea that Sakura plays a role in germ cell maintenance and differentiation through interaction with Otu during oogenesis.Strengths:To my knowledge, this is the first characterization of the role of CG14545 genes. Each experiment seems to be well-designed and adequately controlledWeaknesses:However, the conclusions from each experiment are somewhat separate, and the functional relationships between Sakura's functions are not well established. In other words, although the loss of Sakura in the germline causes pleiotropic effects, the cause-and-effect relationships between the individual defects remain unclear.Comments on latest version:The authors have attempted to address my initial concerns with additional experiments and refutations. Unfortunately, my concerns, especially my specific comments 1-3, remain unaddressed. The present manuscript is descriptive and fails to describe the molecular mechanism by which Sakura exerts its function in the germline. Nevertheless, this reviewer acknowledges that the observed defects in sakura mutant ovaries and the possible physiological significance of the Sakura-Out interaction are worth sharing with the research community, as they may lay the groundwork for future research in functional analysis.

We thank the reviewer for valuable comments. We would like to investigate the molecular mechanism by which Sakura exerts its function in the germline in near future studies.

**Reviewer #2 (Public review):**
In this study, the authors identified CG14545 (named it sakura), as a key gene essential for *Drosophila* oogenesis. Genetic analyses revealed that Sakura is vital for both oogenesis progression and ultimate female fertility, playing a central role in the renewal and differentiation of germ stem cells (GSC).The absence of Sakura disrupts the Dpp/BMP signaling pathway, resulting in abnormal bam gene expression, which impairs GSC differentiation and leads to GSC loss. Additionally, Sakura is critical for maintaining normal levels of piRNAs. Also, the authors convincingly demonstrate that Sakura physically interacts with Otu, identifying the specific domains necessary for this interaction, suggesting a cooperative role in germline regulation. Importantly, the loss of otu produces similar defects to those observed in sakura mutants, highlighting their functional collaboration.The authors provide compelling evidence that Sakura is a critical regulator of germ cell fate, maintenance, and differentiation in *Drosophila*. This regulatory role is mediated through modulation of pMad and Bam expression. However, the phenotypes observed in the germarium appear to stem from reduced pMad levels, which subsequently trigger premature and ectopic expression of Bam. This aberrant Bam expression could lead to increased CycA levels and altered transcriptional regulation, impacting piRNA expression. In this revised manuscript, the authors further investigated whether Sakura affects the function of Orb, a binding partner they identified, in deubiquitinase activity when Orb interacts with Bam.We appreciate the authors' efforts to address all our comments. While these revisions have greatly improved the clarity of certain sections, some of the concerns remain unclear, while details mentioned in the responses about these studies should be incorporated in the manuscript. Specifically, the manuscript still lacks the demonstration that Sakura co-localizes with Orb/Bam despite having the means for staining and visualization. This would bring insight into the selective binding of Orb with Bam vs. Sakura perhaps at different stages of oogenesis. Such analyses would allow for more specific conclusions, further alluding to the underlying mechanism, rather than the general observations currently presented.This elaborate study will be embraced by both germline-focused scientists and the developmental biology community.

We thank the reviewer for valuable comments. We believe that the author meant Otu, not Orb, for the binding partner of Sakura that we identified. We would like to investigate the colocalization of Sakura with other proteins including Otu and the molecular mechanism by which Sakura exerts its function in the germline in near future studies.

**Reviewer #3 (Public review):**
In this very thorough study, the authors characterize the function of a novel *Drosophila* gene, which they name Sakura. They start with the observation that sakura expression is predicted to be highly enriched in the ovary and they generate an anti-sakura antibody, a line with a GFP-tagged sakura transgene, and a sakura null allele to investigate sakura localization and function directly. They confirm the prediction that it is primarily expressed in the ovary and, specifically, that it is expressed in germ cells, and find that about 2/3 of the mutants lack germ cells completely and the remaining have tumorous ovaries. Further investigation reveals that Sakura is required for piRNA-mediated repression of transposons in germ cells. They also find evidence that sakura is important for germ cell specification during development and germline stem cell maintenance during adulthood. However, despite the role of sakura in maintaining germline stem cells, they find that sakura mutant germ cells also fail to differentiate properly such that mutant germline stem cell clones have an increased number of "GSC-like" cells. They attribute this phenotype to a failure in the repression of Bam by dpp signaling. Lastly, they demonstrate that sakura physically interacts with otu and that sakura and otu mutants have similar germ cell phenotypes. Overall, this study helps to advance the field by providing a characterization of a novel gene that is required for oogenesis. The data are generally high-quality and the new lines and reagents they generated will be useful for the field.Comments on latest version:With these revisions, the authors have addressed my main concerns.

We thank the reviewer for valuable comments.

**Recommendations for the authors:**

**Reviewer #2 (Recommendations for the authors):**
The manuscript is much improved based on the changes made upon recommendations from the reviewers.Though most of our comments have been addressed, we have a few more we wish to recommend. For previous points we made, we replied with further clarification for the authors.Figure 1(1) B should be the supplemental figure.We moved the former Fig 1B to Supplemental Figure 1.• Previous Fig1B (sakura mRNA expression level) is now Fig S2, not S1. Please make this data as Fig S1.

We moved Fig S1 to main Fig7A and renumbered Fig S2-S16 to Fig S1-S15.

(2) C - How were the different egg chamber stages selected in the WB? Naming them 'oocytes' is deceiving. Recommend labeling them as 'egg chambers', since an oocyte is claimed to be just the one-cell of that cyst.We changed the labeling to egg chambers.• The labels on lanes for Stages 12-13 and Stage 14, still only say "chambers", not "egg chambers". Also there is no Stage 1-3 egg chamber. More accurately, the label should be "Germarium - Stage 11 egg chambers".

We updated the lables on lanes as suggested by the reviewer.

(3) Is the antibody not detecting Sakura in IF? There is no mention of this anywhere in the manuscript.While our Sakura antibody detects Sakura in IF, it seems to detect some other proteins as well. Since we have Sakura-EGFP fly strain (which fully rescues sakuranull phenotypes) to examine Sakura expression and localization without such non-specific signal issues, we relied on Sakura-EGFP rather than anti-Sakura antibodies for IF.• Please put this info into the Methods section.

We added this info into the Methods section.

(4) Expand on the reliance of the sakura-EGFP fly line. Does this overexpression cause any phenotypes?sakura-EGFP does not cause any phenotypes in the background of sakura[+/+] and sakura[+/-].• Please add this detail into the manuscript.

We added this info into the Methods section.

Figure 5(1) D - It might make more sense if this graph showed % instead of the numbers.We did not understand the reviewer's point. We think using numbers, not %, makes more sense.• Having a different 'n' number for each experiment does not allow one to compare anything except numbers of the egg chambers. This must be normalized.

We still don’t agree with the reviewer. In Fig 5D, we are showing the numbers of stage 14 oocytes per fly (=per a pair of ovaries). ‘n’ is the number of flies (=number of a pair of ovaries) examined. We now clarified this in the figure legend. Different ‘n’ number does not prevent us from comparing the numbers of stage 14 oocytes per fly. Therefore, we would like to show as it is now.

(2) Line 213 - explain why RNAi 2 was chosen when RNAi 1 looks stronger.Fly stock of RNAi line 2 is much healthier than RNAi line 1 (without being driven Gal4) for some reasons. We had a concern that the RNAi line 1 might contain an unwanted genetic background. We chose to use the RNAi 2 line to avoid such an issue.• Please add this information to the manuscript.

We added this info into the Methods section.

Figure 7/8 - can go to Supplemental.We moved Fig 8 to supplemental. However, we think Fig 7 data is important and therefore we would like to present them as a main figure.• Current Fig S1 should go to Fig 7, to better understand the relationship between pMad and Bam expression.

We moved Fig S1 to main Fig7A and renumbered Fig S2-S16 to Fig S1-S15.

Figure 9C - Why the switch to S2 cells? Not able to use the Otu antibody in the IP of ovaries?We can use the Otu antibody in the IP of ovaries. However, in anti-Sakura Western after anti Otu IP, antibody light chain bands of the Otu antibodies overlap with the Sakura band. Therefore, we switched to S2 cells to avoid this issue by using an epitope tag.• Please add this info to the Methods section.

We added this info into the Methods section.

Figure 10- Some images would be nice here to show that the truncations no longer colocalize.We did not understand the reviewer's points. In our study, even for the full-length proteins. We have not shown any colocalization of Sakura and Otu in S2 cells or in ovaries, except that they both are enriched in developing oocytes in egg chambers.• Based on your binding studies, we would expect them to colocalize in the egg chamber, and since there are antibodies and a GFP-line available, it would be important to demonstrate that via visualization.

As we wrote in the response and now in the manuscript, our antibodies are not best for immunostaining. We will try to optimize the experimental conditions in the future studies.